# SPLIT AND MERGE: ALIGNING POSITION BIASES IN LARGE LANGUAGE MODEL BASED EVALUATORS

## ABSTRACT

Large language models (LLMs) have shown promise as automated evaluators for assessing the quality of answers generated by AI systems. However, these LLM-based evaluators exhibit position bias, or inconsistency, when used to evaluate candidate answers in pairwise comparisons, favoring either the first or second answer regardless of content. To address this limitation, we propose PORTIA, an alignment-based system designed to mimic human comparison strategies to calibrate position bias in a lightweight yet effective manner. Specifically, PORTIA splits the answers into multiple segments, aligns similar content across candidate answers, and then merges them back into a single prompt for evaluation by LLMs. We conducted extensive experiments with six diverse LLMs to evaluate 11,520 answer pairs. Our results show that PORTIA markedly enhances the consistency rates for all the models and comparison forms tested, achieving an average relative improvement of 47.46%. Remarkably, PORTIA enables less advanced GPT models to achieve 88% agreement with the state-of-the-art GPT-4 model at just 10% of the cost. Furthermore, it rectifies around 80% of the position bias instances within the GPT-4 model, elevating its consistency rate up to 98%. Subsequent human evaluations indicate that the PORTIA-enhanced GPT-3.5 model can even surpass the standalone GPT-4 in terms of alignment with human evaluators. These findings highlight PORTIA's ability to correct position bias, improve LLM consistency, and boost performance while keeping cost-efficiency. This represents a valuable step toward a more reliable and scalable use of LLMs for automated evaluations across diverse applications.

## 1 INTRODUCTION

Recent advances in large language models (LLMs) have achieved remarkable results on various tasks, sometimes even exceeding human performance (Kojima et al., 2022; Thapa et al., 2023). However, assessing the quality of LLM-generated answers poses challenges. Specifically, n-gram matching metrics like BLEU (Papineni et al., 2002) can quantify token-level overlap with reference texts but fall short in evaluating semantic quality. While human evaluators provide more accurate and valuable feedback, often considered the "gold standards," their scalability is generally low, given that they are costly and time-consuming. As a result, there emerges a growing need for automated evaluation methods that reliably align with human yet remain efficient and cost-effective.

Recently, researchers have investigated the use of powerful LLMs like GPT-4 (OpenAI, 2023) to evaluate the quality of text generated in response to open-ended questions (Zheng et al., 2023). Notably, robust LLM evaluators such as GPT-4 have been shown to align remarkably well with both controlled and crowdsourced human preferences, achieving over 60% agreement (Wang et al., 2023a). These studies suggest that LLMs can emulate human evaluations, offering a scalable and transparent alternative to the expensive and time-intensive human assessment of text quality.

While LLMs have advanced capabilities, they are not flawless evaluators and have been identified to possess certain biases. One notable bias is the position bias (Zheng et al., 2023; Wang et al., 2023a), in which an LLM might prefer either the first or second answer in a pairwise comparison, regardless of its content, as illustrated in Figure 1. Even the state-of-the-art GPT-4 model is not immune to position bias (Zheng et al., 2023; Wang et al., 2023a; Zhang et al., 2023; Zeng et al., 2023), and the behavior of its various versions can be inconsistent over time (Chen et al., 2023).

Moreover, owing to pronounced position biases in less-powerful GPT models, much of the prior research (Zheng et al., 2023; Zhang et al., 2023) has been compelled to use the expensive GPT-4 for LLM evaluations, emphasizing the necessity for a more cost-effective approach to large-scale assessments.

To address these limitations, we propose PORTIA[1], an alignment-based system designed to calibrate position bias. Inspired by human long-text reading strategies (Ratnasari, 2023), PORTIA splits the answers into multiple segments, aligns similar content across candidate answers, and then merges them back into a single prompt to feed to LLM evaluators. Specifically, PORTIA first identifies possible split positions at sentence boundaries within each answer. It then conducts a length alignment between the candidates to generate segments of roughly equal length across answers. If this length alignment does not yield a consistent verdict, PORTIA further undertakes an iterative semantic alignment to identify the optimal split positions, enabling the merging of segments across candidates. Since this lightweight approach does not require changes to the models themselves, PORTIA is readily adaptable to enhance a variety of LLM evaluators for improved evaluation consistency.

We conducted comprehensive experiments using six LLMs as evaluators to assess 11,520 answer pairs across three prevalent pairwise comparison forms. Our results show that PORTIA markedly boosts consistency rates for all the tested models and templates, achieving an average relative improvement of 47.46% and rectifying an average of 62.31% of the initially inconsistent cases. Furthermore, PORTIA addresses between 36% and 86% (over 80% for two-thirds of the comparison templates) of the position bias occurrences within the GPT-4 model, elevating its consistency rate up to 98%. Moreover, efficiency and cost evaluations indicate that PORTIA enables the less advanced GPT-3.5 model to achieve 88% agreement with the state-of-the-art GPT-4 model at merely 9.57% of the cost. Additionally, a user study involving five human participants demonstrated enhanced agreement between PORTIA-optimized evaluators and human evaluators. Remarkably, the agreement of human evaluators with PORTIA-enhanced GPT-3.5 even exceeds that with the standalone GPT-4. A subsequent ablation study suggests that PORTIA's two key components — length alignment and semantic alignment — are beneficial for improving consistency across different comparison forms.

## 2 BACKGROUND

**Paradigms of Using LLM-based Evaluators.** Recent work has explored using LLMs such as GPT-4 to evaluate and compare the performance of AI systems (Wang et al., 2023a; Chan et al., 2023; Zheng et al., 2023; Hada et al., 2023). Conceptually, there are two distinct LLM-based comparison paradigms: *single-wise comparison* and *pairwise comparison*. In single-wise comparison, LLM evaluators are provided with one answer each time and are asked to score each answer independently, causing that position bias is not an issue in single-wise LLM evaluation and therefore beyond the scope of this paper. Nevertheless, we find that the absolute scores of LLM may lack clear interpretation. To demonstrate this, we conducted a preliminary study where we examined the consistency of single-wise comparison across a total of 80 test cases, each involving three sets of value ranges. Our findings indicate that the scores from single-wise comparison do not strictly adhere to a linear mapping relationship across different scales (more discussion in Appendix B).

Pairwise comparison presents two answers side-by-side and asks evaluators to select the superior one. In particular, pairwise comparison methods can be further categorized into three forms: *score-based*, *likert-based*, and *relation-based*. In score-based comparison, evaluators assign a score to each answer and then compare these scores to determine the better answer. The likert-based method (Rajani et al., 2023) requires evaluators to score answers on a likert scale (Likert, 1932), where lower scores indicate a strong preference towards the first answer, middle scores represent a close tie, and higher scores signal a preference for the second answer. Additionally, the relation-based comparison solicits direct inputs from the evaluators about their preference for one answer over another. This approach aims to avoid the use of potentially arbitrary scores, guiding evaluators to make relative comparisons between answers instead. The details of these three forms are shown in Appendix A.1.

---

[1]The name PORTIA is inspired by the intelligent and astute character, Portia, from Shakespeare's "The Merchant of Venice." In the play, Portia assists a judge in making fair decisions within the legal rules. Just as Portia requests the exact amount of flesh to be cut, our method seeks to make fair splits of the original answers for comparison.

**Position Bias in Pairwise Comparison.** Despite the generally encouraging performance of pairwise comparison methods, we note that LLM evaluators are not perfect and can exhibit certain biases. A primary concern is the *position bias* (Zheng et al., 2023; Wang et al., 2023a), whereby the LLM may favor the first (or second) answer in a pairwise comparison, regardless of its content. In fact, LLMs have shown notable sensitivity to small changes in prompts (Zhao et al., 2021; Zhu et al., 2023). For clarity, we provide a formal definition of position bias as well as the consistency. As illustrated in Table 4 (refer to Appendix A.1), the evaluation input comprises a fixed template with three placeholders. The input set for the LLM evaluators can be represented as $\{Q, R_1, R_2\}$, where $Q$ denotes the question set, and $R_1$ and $R_2$ are the two sets of answers for comparison. The LLM evaluators produce the verdict $V = LLM(\{Q, R_1, R_2\})$, which indicates the preferred answer out of the two candidates. Assuming that the LLM evaluators are flawless, the verdict $V$ should be independent of the permutation $\Pi$ of $R_1$ and $R_2$. Thus, position bias can be expressed as: $\Pi \not\perp V$. On an individual sample level, for a specific question $q \in Q$ and answers $r_1$, $r_2$, consistency is achieved if the verdict $v$ remains the same when the positions of $r_1$, $r_2$ are switched: $LLM(\{q, r_1, r_2\}) = LLM(\{q, r_2, r_1\})$.

## 3 THE PORTIA SYSTEM

**Design Intuition.** It is worth noting that both human evaluators and LLMs encounter difficulties in making consistent evaluations when faced with lengthy and intricate answers (Kintsch & Keenan, 1973; Wijesiriwardene et al., 2023). A common cognitive approach among individuals is to decompose information into smaller units, thereby simplifying the comparison process (Ratnasari, 2023). Inspired by this observation, PORTIA is designed to split candidate answers into segments, merge specific segments across candidates that share "comparable" content, and eventually align them. Based on this intuition, PORTIA seeks to mimic effective human comparison procedures, aiming to calibrate position bias and enhance the consistency of LLM evaluators. That said, for each question, the verdicts of PORTIA should be consistent with any permutation of the answers, i.e., $LLM(\{q, r_1, r_2\}) = LLM(\{q, r_2, r_1\})$.

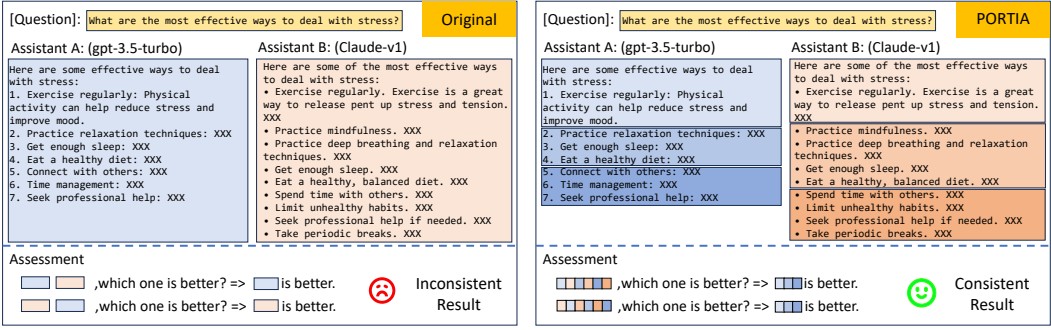

Figure 1: A sample pairwise LLM-based evaluation improved by PORTIA. *Left:* The original evaluation exhibiting inconsistency. *Right:* Consistent evaluation after applying PORTIA. Details of the answers, comparison forms, and evaluation contents have been simplified or omitted for clarity. For more detailed information, readers can refer to Appendix A.2.

### 3.1 CONSIDERATIONS IN DESIGNING A "SPLITTING & MERGING" SCHEME

Prior to presenting the technical design of PORTIA, it is imperative to introduce the following key design considerations. It should be noted that our objective is *not* to improve the original answers, but rather to assist the evaluators in accurately evaluating the quality of the answers. Consequently, our design objectives encompass the attainment of three specific properties:

**Content Preservation.** Content preservation refers to ensuring the segmented answers encompass the entirety of the information present in the original answer, without any omissions or additions of new content. For a given original answer $r_1$, the set of split answer segments $\{r_1^1, r_1^2, ..., r_1^k\}$ should fully encompass the content of $r_1$. This implies that when the segments are concatenated, the entirety of the original content is preserved ($\sum_{i=1}^{k} r_1^i = r_1$). This consideration helps to preserve the meaning and information of the original answer during the process of splitting. The preservation of content is critical in order for evaluators to assess the same substantive answer content that is divided into segments, without any alterations or incomplete information.

**Order Preservation.** Order preservation refers to preserving the original sequence of the information presented in the answer. This is important for fair evaluation, as re-ordering or re-arranging the content may impact the assessment of answer quality. For example, if the sequence of actions (i.e., answer) in response to the question "What should you do in the morning after waking up?" is re-ordered such that "eat lunch" before "brush teeth," it may be perceived as an answer of lower quality. By preserving the order, we ensure the segmentation process does not introduce artifacts that could unintentionally alter assessment. This enables the LLM evaluators to accurately evaluate answers in comparison to the original.

**Resource efficiency.** Resource efficiency refers to minimizing computational costs incurred by the splitting process, beyond the standard cost when querying the LLM evaluator. To this end, it is important for the segmentation process to introduce a minimal number of extra tokens and to be executed rapidly, thus avoiding significant overhead.

## 3.2 THE CORE SPLITTING ALGORITHM

Due to the page limit, we direct interested readers to Appendix C for a comprehensive overview of utilizing PORTIA for LLM evaluation. Here we concentrate on PORTIA's core splitting algorithm, as illustrated in Alg. 1. Intuitively, PORTIA first identifies semantically or syntactically similar segments across answers. It then aligns these answer segments and merges them sequentially into a single prompt for the LLM evaluators to make a final verdict. Specifically, the inputs include the question $q$, two candidate answers $r_1$ and $r_2$, the LLM evaluator's verdict function $v()$, and the specified number of splits $k$. The output of Alg. 1 is a consistent verdict $v \in (1, 2, 3)$, where 1 indicates that $r_1$ is superior, 2 suggests that $r_2$ is better, and 3 represents a tie.

---

**Algorithm 1:** Alignment-based Splitting Process (PORTIA)

**Input:** Question: $q$, Answers: $r_1, r_2$, Evaluator's verdict $v()$, Split number $k$
**Output:** Consistent evaluation $v \in (1, 2, 3)$

```
/* Step1:  identify answers' formats with split positions.          */
```
1   $r_1^{positions} = format(r_1), r_2^{positions} = format(r_2)$
```
/* Step2:  length alignment.                                         */
```
2   $[r_1^{(1)}, ...r_1^{(k)}] = equalsplit(r_1^{positions}, k), [r_2^{(1)}, ...r_2^{(k)}] = equalsplit(r_2^{positions}, k)$
3   **if** $v(q_i, r_1^{(1)}, r_2^{(1)}, ..., r_1^{(k)}, r_2^{(k)}) == v(q_i, r_2^{(1)}, r_1^{(1)}, ..., r_2^{(k)}, r_1^{(k)})$ **then**
```
    /* Consistent, return answer                                     */
```
4      **return** $v$
5   **end**
```
/* Step3:  semantic alignment.                                       */
```
6   **else**
7      $s_{max} = 0, n_s = 0, Search\_all = False, r_1^{bestparts} = [], r_2^{bestparts} = []$
8      **while** *not Search_all* **do**
9          $r_1^{parts} = partition(r_1^{positions}, k, n_s), r_2^{parts} = partition(r_2^{positions}, k, n_s), n_s += 1$
10          $s_{cum} = \sum_{i=1}^{k} similarity(r_1^{parts}[i], r_2^{parts}[i])$
```
        /* Update max similarity score, keep best split positions.   */
```
11          **if** $s_{cum} > s_{max}$ **then**
12             $s_{max} = s_{cum}, r_1^{bestparts} = r_1^{parts}, r_2^{bestparts} = r_2^{parts}$
13          **end**
14      **end**
15      **if** $v(q_i, r_1^{(1)}, r_2^{(1)}, ..., r_1^{(k)}, r_2^{(k)}) == v(q_i, r_2^{(1)}, r_1^{(1)}, ..., r_2^{(k)}, r_1^{(k)})$ **then**
16          **return** $v$
17      **end**
18 **end**

---

Overall, the splitting process can be divided into three stages. In the first phase, possible split positions are determined at the boundaries of sentences (line 1). Segmenting at sentence breaks (e.g., periods or question marks) reduces the likelihood of producing incomplete words or fragmented syntactic units in different segments. This particular design decision aids in maintaining semantic consistency and enhancing readability in each segment. Notably, natural language and programming language have different definitions for sentence boundaries; for instance, the period sign "." in Python denotes accessing a specific object member property. Therefore, in instances where answers

involve code blocks, we leverage `treesitter` (tre) to parse code blocks and locate suitable split positions that preserve the code's structure and execution sequence. This allows PORTIA to split lengthy pieces of code into smaller, logically connected units to facilitate more accurate comparison.

The second stage performs length alignment, splitting each answer into $k$ segments of comparable length (line 2). Specifically, we first find the $k-1$ points that divide the answer into $k$ equal segments according to the number of characters. Subsequently, we select the split location that is closest to each of the split positions obtained in the first stage, and designate them as $[r_1^{(1)}, ... r_1^{(k)}]$.[2] The $k$ corresponding answer segments are subsequently merged again and used for evaluation by the LLM evaluator. If the LLM evaluator consistently returns the same verdicts for all length-aligned splits, then the verdict is returned (lines 3-5).

If inconsistent assessments persist after length alignment, PORTIA proceeds to semantic alignment as the third stage (lines 7-14). Specifically, given a fixed $k$ and a set of possible split positions, we aim to iteratively search for the optimal split positions that maximize the cumulative semantic similarity between corresponding segments of the two answers. Note that $n_s$ represents the index number of the current segmentation, and $Search\_all$ becomes `True` when $n_s$ reaches the maximum number of possible split combinations $Cal$. Semantic similarity between segments $r_1^t$ and $r_2^t$ is computed by token overlap: $sim\_score = \frac{Intersection(set(r_1^t), set(r_2^t))}{\max(\text{len}(set(r_1^t)), \text{len}(set(r_2^t)))}$. Notably, the choice of value $k$ as well as the similarity metric would have an impact on the efficiency of PORTIA, and we provide the theoretical analysis in Section 4.3. We also consider applying other similarity metrics, such as LM-based metrics (Reimers & Gurevych, 2019). However, we argue that employing such intricate metrics is not necessary for PORTIA, as they usually entail extra computing resources, and introduce more hyper-parameters while yielding only marginal improvements in performance; see further discussion in Appendix E. Finally, PORTIA would yield consistent verdict if applicable (lines 15-17). Note that the above three stages are carried out in a sequential manner, whereas semantic alignment is only performed when length alignment is inadequate for ensuring consistent assessments.

# 4 EXPERIMENTS

## 4.1 EXPERIMENTAL SETUP

**Datasets.** We evaluate PORTIA using the MT-Bench benchmark (Zheng et al., 2023), following the experimental setup in Wang et al. (2023a). MT-Bench contains 80 elaborated questions spanning 8 categories (Writing, Roleplay, Reasoning, Math, Coding, Extraction, STEM, and Humanities). For each question, MT-Bench provides several candidate answers from different LLMs. We consider eight different combinations of LLM answers (see more details in Appendix D), and we consider all three comparison forms (score-based, likert-based, and relation-based) in the pairwise comparison paradigm. Thus, we have $80 * 8 * 3 = 1920$ inputs to evaluate each LLM evaluator. We interpret the datasets as large and diverse enough to provide a comprehensive evaluation of PORTIA across different LLMs and comparison forms.

**Implementation Details.** In this work, we include both locally deployable models that are open-source and proprietary models that are accessed through only cloud APIs as LLM evaluators. For local models, we select Chatglm2 (Zeng et al., 2022) and Llama2 (Touvron et al., 2023), due to their notable efficacy and convenient local deployment capabilities. For cloud-based LLMs, we use GPT (including both GPT-4 and GPT-3.5) (OpenAI, 2023) from OpenAI, Qwen (qwe) from Alibaba, and Claude2 (cla) from Anthropic. The rationale for using these models is based on their exceptional performance, since they are considered among the most advanced and powerful in the world. Details on the specific LLM versions evaluated are provided in Appendix D. We run experiments on a GPU server with Intel Xeon Platinum 8276 CPU, 256GB of RAM, and 4 NVIDIA A100 GPUs. This server is capable of performing cloud API calls and local LLM inference.

**Deterministic Results.** To assure reproducibility, we employ various methods to mitigate the inherent randomness in the decoding process of LLMs. For models using cloud API, the hyper-parameter "temperature" is uniformly set to 0 across all evaluators. For local models, the sampling function is deactivated during the decoding phase to get deterministic results.

---

[2]We present an illustration with two detailed algorithms in Appendix F to ease the understanding.

| Evaluators | De. Method | Model | Relation-based | Score-based | Likert-based |
|---|---|---|---|---|---|
| Claude2 | API | % Origin Con
% PORTIA Con
% Fixed Coverage | 28.28
**83.28** (↑194.48%)
79.44 | 47.34
**65.16** (↑37.64%)
52.22 | 50.62
**94.84** (↑87.36%)
91.27 |
| Qwen | API | % Origin Con
% PORTIA Con
% Fixed Coverage | 63.12
**78.13** (↑23.78%)
65.66 | 52.66
**71.09** (↑35.0%)
59.78 | 8.12
**9.38** (↑15.52%)
6.46 |
| Chatglm2 | Local | % Origin Con
% PORTIA Con
% Fixed Coverage | 38.44
**61.72** (↑60.56%)
56.09 | 58.59
**74.06** (↑26.4%)
51.02 | 26.72
**64.22** (↑140.34%)
60.30 |
| Llama2 | Local | % Origin Con
% PORTIA Con
% Fixed Coverage | 36.41
**68.75** (↑88.82%)
22.51 | N/A
**N/A**
N/A | N/A
**N/A**
N/A |
| GPT-3.5 | API | % Origin Con
% PORTIA Con
% Fixed Coverage | 78.12
**88.59** (↑13.4%)
70.63 | 39.22
**54.84** (↑39.83%)
42.06 | 78.91
**98.60** (↑24.94%)
96.32 |
| GPT-4 | API | % Origin Con
% PORTIA Con
% Fixed Coverage | 93.44
**97.03** (↑3.84%)
80.99 | 92.75
**98.00** (↑5.66%)
86.33 | 61.50
**63.50** (↑3.25%)
36.09 |

Table 1: The main results of PORTIA across LLM evaluators. All metrics presented are higher-is-better values. "% Origin Con" and "% PORTIA Con" are the percentages of consistent results in the original setting when enhanced by PORTIA, respectively. "% Fixed Coverage" denotes the percentage of inconsistent original assessments that are later corrected by PORTIA. "De Method" specifies whether the LLM evaluator uses local or cloud API deployment.

## 4.2 MAIN RESULTS

As shown in Table 1, PORTIA improves the consistent rate among all evaluators. The values depicted in the table correspond to the mean values obtained from the analysis of all eight combinations of tested models. We observe that PORTIA relatively improves the consistent rate by 3.25% to 194.48%, depending on the evaluator, with the highest fixed coverage at 96.32% (meaning that nearly all the inconsistent results are resolved). GPT-4 exhibits the highest average consistency rate, which is in line with the findings of previous work (Wang et al., 2023a), and PORTIA further boosts its consistency up to 98%. Moreover, we observe that GPT-4 exhibits subpar performance on the likert-based form, not just compared to its performance on other forms, but also when compared to GPT-3.5. Upon analyzing results on likert-based forms, over 78% of GPT-4's inconsistency provides a score of 5, reflecting its bias for the second answer, and our method rectifies 36.09% of them. Notably, we only report the results of Llama2 in relation-based form, as it fails to provide meaningful evaluations in score-based and likert-based forms (see more details in Appendix G).

The impact of the comparison form on consistency rates is also observed, with evaluators displaying various preferences. For instance, it is seen that GPT-3.5 exhibits the least consistent performance when evaluated on the score-based form, whereas Claude2 struggles most on the relation-based form. GPT-4, Qwen, and Chatglm2 exhibit the highest degree of inconsistency when assessed on the likert-based form. This suggests that appropriately matching comparison forms to evaluators' capabilities is important. Nevertheless, PORTIA offers high enhancement for forms and LLM evaluators. The substantial improvements highlight the generalizability of PORTIA. In summary, these findings clearly validate PORTIA's effectiveness at mitigating inconsistency for both cutting-edge and less powerful LLM evaluators.

## 4.3 EFFICIENCY AND COST ANALYSIS

To demonstrate the efficiency and cost-effectiveness of PORTIA, this section first performs a theoretical analysis of PORTIA's efficiency, and then evaluates its actual costs in terms of temporal, monetary, and environmental factors. Specifically, we measure the efficacy of PORTIA-enhanced LLMs in terms of their agreement rate with GPT-4.

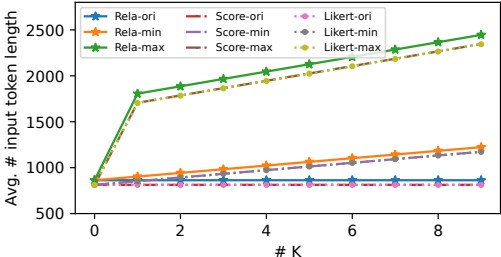
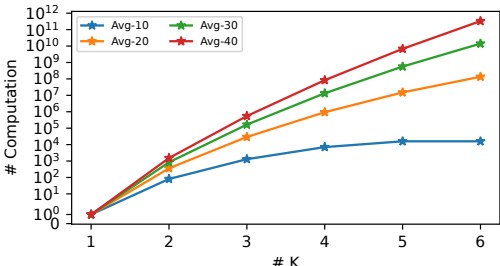

(a) Average input token length with different $k$. "ori" represents the original input length. "min" and "max" represent the minimum and maximum input lengths, respectively.

(b) Computation operations with different $k$. The number behind "Avg" is the average potential split positions for each answer. The y-axis has been logarized to ease reading.

Figure 2: Theoretical estimation of PORTIA's cost with different $k$ in terms of input length and computation operations.

**Theoretical analysis.** We first theoretically analyze PORTIA's computational efficiency. As the number of answer segments $k$ increases, the average input length for LLM evaluators also grows correspondingly. In line with line 1 in Alg. 1, the added tokens stem from two sources: fixed-length system prompts based on comparison forms, and split boundary prompts (an example shown in Table A.2) that scale linearly with $k$. Consequently, the additional input length scales as $O(K)$, as depicted in Figure 2(a). Notably, the average input length of the relation-based form exceeds the other two, as it requires more tokens for the system prompt. More details are in Appendix A.

In accordance with line 9 in Alg. 1, the total number of computation operations $Cal$ is calculated as: $Cal = C_{p_1}^{k-1} * C_{p_2}^{k-1}$, where $p_1$ and $p_2$ are the potential split positions in the two answers. $C_{p_1}^{k-1}$ and $C_{p_2}^{k-1}$ are the combination counts for the first and second answers, respectively. Using average position numbers of 10, 20, 30, and 40, we derive the total calculations as depicted in Figure 2(b). Intuitively, raising the value of $k$ can improve the algorithm's performance by exploring more split position combinations. However, this also results in an exponential surge in the total computation operations, compromising efficiency. As such, we conducted controlled experiments to identify the optimal value of $k$, and in our case, we found that setting $k = 3$ strikes a balance between efficiency and precision. Full details about this controlled experiment can be found in Appendix E.

|  | AR origin (%) | AR fix (%) | Carbon Emitted (CO$_2$eq / per 1k) | Avg Cost (USD / per 1k) | Avg Time (s / per 1k) |
|---|---|---|---|---|---|
| GPT-4 | - | - | N/A | 29.78 | 13,446 |
| GPT-3.5 | 82.50 | 88.59 | 7.22 | 2.85 | 2,192 |
| Qwen | 60.83 | 69.58 | N/A | 35.49 | 6,083 |
| Chatglm2 | 20.34 | 39.16 | 2.15 | 4.09 | 1,983 |
| Claude2 | 43.44 | 75.09 | N/A | 27.17 | 11,561 |

Table 2: Real-world comparison of different LLM evaluators' results before and after fix by PORTIA with that of GPT-4, including resource consumption. "AR" denotes the agreement rate with GPT-4.

**Real-World Performance and Cost Analysis.** Next, we measure the level of agreement between the PORTIA-enhanced LLM evaluators and GPT-4 (considered as the "gold standard."). Note that to offer a fair evaluation, we exclusively consider GPT-4 evaluation outputs that are originally consistent. In the context of a question with two possible answers, it is deemed as an agreement only when both GPT-4 and PORTIA-enhanced assessments are consistent and identical. As evidenced in Table 2, agreement rates are enhanced by an average of 16.32% after alignment. Claude2 has the highest gain at 31.65%, while GPT-3.5 achieves the highest agreement rate with GPT-4 at 88.59%.

Additionally, we take consideration of the resource usage in terms of temporal, monetary, and environmental factors. As shown in Table 2, Chatglm2 exhibits the lowest inferencing time. However, the cost of GPT-3.5 is lower than that of Chatglm2, while its carbon emission is higher[3], which is

---

[3]The carbon emission of GPT-3.5 is estimated following Chien et al. (2023).

mainly due to the fact that the cloud API models usually run on GPU clusters with more powerful GPUs. We estimate the cost using the official pricing for cloud APIs and the Azure ND A100 v4 instances for local models. It is worth mentioning that GPT-3.5 incurs less than **10%** of the average cost of GPT-4, while maintaining an approximate agreement level of 88% with GPT-4. In brief, the usage of PORTIA results in a substantial level of concurrence with GPT-4 while maintaining a minimal computational burden, hence showcasing a proficient and eco-friendly alignment. The significant enhancements in performance and resource utilization underscore the usefulness of this approach in boosting various LLMs for this crucial evaluation work.

## 4.4 HUMAN STUDY

|  | GPT-3.5 | Qwen | Chatglm2 | Claude2 | GPT-4 |
|---|---|---|---|---|---|
| Ori Human AR (%) | 55.00 | 35.00 | 16.25 | 6.25 | 60.00 |
| Fix Human AR (%) | 63.75 | 35.00 | 17.50 | 47.50 | 65.00 |

Table 3: Main results from human evaluation comparing the model pair "gpt-3.5-turbo" v.s. "Claude-v1" on 80 questions. "AR" represents the agreement rate.

We conducted a human evaluation to further assess the performance of PORTIA. The model pair "gpt-3.5-turbo" v.s. "Claude-v1" is selected to compare human agreement rates on original versus PORTIA-enhanced assessments across 80 questions, as these two models have similar performance (Zheng et al., 2023), making it challenging for LLM evaluators to make decisions. We recruit five experts, including two industrial developers and three academic researchers as participants. For each participant, we create an online questionnaire that provides one question with two answers, not specifying their origin. Before the questionnaire, brief instructions on the task and evaluation criteria are provided. During the human evaluation process, we observe some instances where human evaluators make directly opposing assessments. This highlights the inherent subjectivity and unreliability of human evaluation. We attribute these disagreements to the diversity of human values (Peng et al., 1997), and simply use a majority vote to determine the final result.

The human evaluation results presented in Table 3 demonstrate increased agreement rates between humans and LLM evaluators after applying PORTIA. On average, human agreement on original LLM assessments improves by 11.25% after enhancement. Notably, the original human agreement rate for Claude2 is only 6.25%, but increases substantially to 47.50% after enhancement. In addition, while the original human agreement lags behind GPT-4 across evaluators, PORTIA-enhanced GPT-3.5 surpasses the original GPT-4, indicating enhanced consensus. Taken together, these quantitative findings provide evidence that PORTIA effectively augments the assessments of all LLM evaluators to achieve greater concordance with human evaluators. The framework also enables weaker LLMs to reach comparability with stronger counterparts in terms of human alignment.

## 4.5 ABLATION STUDY

To ascertain the individual contributions of each component in PORTIA, we conduct ablation experiments on five distinct LLM evaluators. The results are quantified in terms of the "Fixed Coverage" metric, as depicted in Figure 3. To facilitate visual interpretation, variants of PORTIA incorporating or excluding specific components are denoted by different colored bars in the histogram. Additionally, texture patterns in the bars indicate the comparison form used. The plain blue bar represents the score-based form, while the blue bar with slash lines corresponds to the relation-based form.

The results reveal that both semantic and length alignment confer improvements to PORTIA's performance. Specifically, across all evaluators, semantic alignment shows a greater contribution to enhancing the likert-based form, which is likely attributable to the likert scale's greater dependence on precise semantic meaning for its standardized categorical ratings. In contrast, for the score-based and relation-based forms, both alignment methods contribute comparably, with slight differences between the LLM evaluators. A plausible explanation is that the latter two forms better imitate human evaluators by considering semantic meaning and answer length in a balanced way.

Furthermore, we find that the trends of fixed coverage rate are consistent across comparison forms for PORTIA and ablations (without semantic or length alignment). likert-based form has the highest fixed coverage rates, followed by relation-based, with score-based form having the lowest rates. The

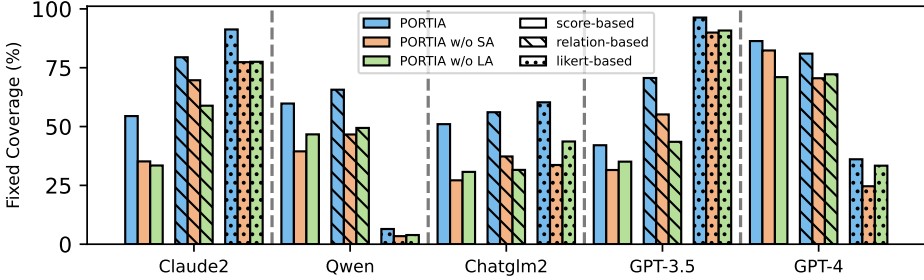

Figure 3: Fixed coverage rate across five LLM evaluators for PORTIA and variants excluding semantic alignment (PORTIA w/o SA) or length alignment (PORTIA w/o LA).

exceptions are Qwen on likert-based form and GPT-4 on all forms, where we manually check and find that: (1) Qwen prefers the second answer for over 90% of examples, no matter whether PORTIA is used. (2) GPT-4 has the highest fixed coverage rate on relation-based form, which is probably because GPT-4 performs consistently enough (more than 97% consistent rate), and therefore, the improvement on likert-based form is not obvious. Overall, aside from the outliers, these results suggest that likert-based form is the most challenging, and we attribute this to that it requires the evaluators to assign a single score that contains an assessment of two answers, which is more difficult than simply choosing the better one like relation-based form.

## 5 RELATED WORK

**Automatic Evaluation of AI Outputs.** Automated evaluation using standard metrics like BLEU (Papineni et al., 2002) and ROUGE (Lin, 2004) is a popular approach to assessing the quality of AI-generated text. However, these metrics are limited in their ability to assess meaning, reasoning, and logical consistency. Recent efforts have focused on developing more robust semantic similarity metrics using neural representations (Zhang et al., 2019), but they are still imperfect proxies for human assessment. LLM has emerged as a promising alternative for evaluation. Notably, Chiang & yi Lee (2023) were the first to demonstrate the potential of LLMs as an alternative to human evaluation. Meanwhile, G-EVAL (Liu et al., 2023) designed a prompt-based evaluator to assess the quality of natural language generation outputs. LLM-EVAL (Lin & Chen, 2023) provided a unified schema for evaluating multiple dimensions of conversation quality, while PandaLM (Wang et al., 2023b) enabled reproducible automated assessment by collecting diverse human annotations and training a model to predict fair assessments.

**Biases in LLM Evaluators.** Besides position bias, Zheng et al. (2023) identify two additional biases: verbosity bias, which refers to a preference for longer answers, and self-enhancement bias, which involves a preference for self-generated answers. However, the definition of verbosity bias is not clear, and we observe that human evaluators also tend to prefer longer answers. Furthermore, self-enhancement bias is not universal for all evaluators (Zheng et al., 2023). Given these considerations, we focus on addressing position bias, as its mitigation can directly improve the efficiency and accuracy of a wide variety of LLM evaluators already in real-world use (Li et al., 2023).

**Multi-agent LLMs.** In addition to single-agent LLM, researchers have explored multi-agent LLMs. For example, Wu & Aji (2023) propose rounds of proposal and debate among multiple LLM instances to reach consensus answers with improved reasoning. Complementarily, Chateval (Chan et al., 2023) implements a multi-agent debate framework to move beyond individual prompting strategies, and the multi-Elo rating system (Wu & Aji, 2023) substantially improves evaluation quality and factual accuracy for multi-LLMs. While powerful, these efforts are orthogonal to our work.

## 6 CONCLUSION

In this paper, we presented PORTIA, an alignment-based technique to address position bias for LLM evaluators. By aligning similar content segments across candidate answers, PORTIA effectively reduced position bias, boosting consistency rates with a relative improvement of 47.46%. Notably, it not only enabled replacing expensive models like GPT-4 with more affordable alternatives but also elevated the consistency rate of GPT-4 itself. PORTIA provided a valuable step towards accessible, eco-friendly LLM evaluators that are more reliable and robust for diverse real-world applications.

**Ethics Statement.** Our work aims to improve the consistency of LLM-based evaluators, which can be utilized to assess the quality of AI-generated answers. Mitigating positional biases in LLM evaluators constitutes an initial step toward addressing higher-level biases in AI systems, including gender and racial biases. More consistent LLM-based evaluators can provide human-like evaluations at a lower cost, supplying feedback to reduce biases during training. However, we recognize that malicious actors could exploit these methods to intentionally train models that go against human values. The open-source LLMs could be leveraged as consistent evaluators to guide the training of harmful models such as Worm-GPT (wor). While our work targets constructive applications, we caution that like any technology, consistent LLM evaluators could potentially be misused. Researchers should consider ethical implications and preventative measures. Overall, we believe the benefits of more fair and accurate AI outweigh the risks, but responsibility is required in deployment.

**Reproducibility Statement.** All our results are reproducible using the code repository we will release. All experimental details, including hyperparameters, are reported in Section 3.2 and Appendix E. We reuse the benchmark datasets from Zheng et al. (2023), with the different comparison prompt forms detailed in Appendix A.

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

# A  PROMPT TEMPLATES

## A.1  COMPARISON FORMS

[Question] {Q}
[The Start of Assistant A's response] {R1} [The End of Assistant A's response]
[The Start of Assistant B's response] {R2} [The End of Assistant B's response]
[System]
Please act as an impartial judge and evaluate the quality of the responses provided by two AI assistants to the user question displayed below.
You should choose the assistant that follows the user's instructions and answers the user's question better. Your evaluation should consider factors such as the helpfulness, relevance, accuracy, depth, creativity, and level of detail of their responses. Begin your evaluation by comparing the two responses and provide a short explanation.
Avoid any positional biases and ensure that the order in which the responses were presented does not influence your decision. Do not allow the length of the responses to influence your evaluation. Do not favor certain names of the assistants. Be as objective as possible.
After providing your explanation, output your final verdict by strictly following this format: [[A]] if assistant A is better, [[B]] if assistant B is better, and [[C]] for a tie.

Table 4: The relation-based evaluation form with three slots ({Q}, {R1} and {R2}) from (Zheng et al., 2023).

[Question] {Q}
[The Start of Assistant A's response] {R1} [The End of Assistant A's response]
[The Start of Assistant B's response] {R2} [The End of Assistant B's response]
[System]
We would like to request your feedback on the performance of two AI assistants in response to the user question displayed above.
Please rate the helpfulness, relevance, accuracy, level of details of their responses. Each assistant receives an overall score on a scale of 1 to 10, where a higher score indicates better overall performance. Please first output a single line containing only two values indicating the scores for Assistant A and B, respectively. The two scores are separated by a space. In the subsequent line, please provide a comprehensive explanation of your evaluation, avoiding any potential bias and ensuring that the order in which the responses were presented does not affect your judgment.

Table 5: The score-based evaluation form with three slots ({Q}, {R1} and {R2}).

[Question] {Q}
[The Start of Assistant A's response] {R1} [The End of Assistant A's response]
[The Start of Assistant B's response] {R2} [The End of Assistant B's response]
[System]
We would like to request your feedback on the performance of two AI assistants in response to the user question displayed above.
Please compare the helpfulness, relevance, accuracy, level of details of their responses.
The rating should be from the set of 1, 2, 3, 4, 5, 6, or 7, where higher numbers indicated that Assistant A was better than Assistant B. Please first output a single line containing only one value indicating the preference between Assistant A and B.
In the subsequent line, please provide a brief explanation of your evaluation, avoiding any potential bias and ensuring that the order in which the responses were presented does not affect your judgment.

Table 6: The likert-based evaluation form with three slots ({Q}, {R1} and {R2}).

[Question] {What are the most effective ways to deal with stress?}
[The Start of Assistant A's response part 1]
Here are some effective ways to deal with stress:
1. Exercise regularly: Physical activity can help reduce stress and improve mood.
[The End of Assistant A's response part 1]
[The Start of Assistant B's response part 1]
Here are some of the most effective ways to deal with stress:
• Exercise regularly. Exercise is a great way to release pent up stress and tension. Even taking a 30 minute walk can help clear your mind and relieve stress.
[The End of Assistant B's response part 1]
[The Start of Assistant A's response part 2]
2. Practice relaxation techniques: Deep breathing, meditation, and yoga are some effective ways to relax and reduce stress levels.
3. Get enough sleep: Lack of sleep can increase stress levels. Try to get at least 7-8 hours of sleep each night.
4. Eat a healthy diet: A well-balanced diet can help reduce stress levels and improve overall health.
[The End of Assistant A's response part 2]
[The Start of Assistant B's response part 2]
• Practice deep breathing and relaxation techniques. Things like meditation, yoga, and mindfulness exercises can help lower your stress levels. Taking some deep breaths can help you calm down in the moment.
• Get enough sleep. Aim for 7 to 8 hours of sleep per night. Lack of sleep can intensify stress. Sleep helps your body and mind rest and recharge.
• Eat a healthy, balanced diet. A healthy diet can boost your mood and reduce stress. Focus on whole foods like fruits, vegetables, whole grains, and lean protein. Limit caffeine and alcohol intake.
• Spend time with others. Social support from family and friends can help reduce stress levels. Connecting to others can help take your mind off your worries and boost your mood.
[The End of Assistant B's response part 2]
[The Start of Assistant A's response part 3]
5. Connect with others: Spending time with friends and family, joining a support group, or participating in social activities can help reduce stress levels and improve mood.
6. Time management: Prioritizing tasks and creating a schedule can help reduce stress and increase productivity.
7. Seek professional help: If stress is impacting your daily life, consider seeking professional help such as therapy or counseling.
[The End of Assistant A's response part 3]
[The Start of Assistant B's response part 3]
• Limit unhealthy habits. Things like smoking, excessive drinking, or drug use can worsen symptoms of stress. Try to avoid these unhealthy habits as much as possible.
• Practice mindfulness. Spending time each day being fully present and aware of your thoughts and feelings can help reduce stress and increase feelings of calm. Try yoga, meditation, or simply sit in a quiet space for a few minutes.
• Seek professional help if needed. If you are dealing with chronic or severe stress, it may help to talk to a therapist. A counselor can provide advice and coping strategies tailored to your situation. Therapy and counseling can be very effective at reducing stress and anxiety.
• Take periodic breaks. It is important to take time each day to unwind and do something you enjoy. Read a book, take up a hobby, or simply relax in a bath. Taking breaks prevents burnout and recharges your body and mind.
[The End of Assistant B's response part 3]
[System]
Please act as an impartial judge and evaluate the quality of the responses provided by two AI assistants to the user question displayed below.
You should choose the assistant that follows the user's instructions and answers the user's question better. Your evaluation should consider factors such as the helpfulness, relevance, accuracy, depth, creativity, and level of detail of their responses. Begin your evaluation by comparing the two responses and provide a short explanation.
Avoid any positional biases and ensure that the order in which the responses were presented does not influence your decision. Do not allow the length of the responses to influence your evaluation. Do not favor certain names of the assistants. Be as objective as possible.
After providing your explanation, output your final verdict by strictly following this format: [[A]] if assistant A is better, [[B]] if assistant B is better, and [[C]] for a tie.

Table 7: The detailed prompt illustrated in Figure 1. We use relation-based form to construct the system prompt. The prompt in green is the "split boundary prompts".

A.2  ALIGNMENT TEMPLATES

# B  A PRELIMINARY STUDY OF SINGLE-WISE COMPARISON

---

[Question] {Q}
[The Start of Assistant A's response] {R1} [The End of Assistant A's response]
[The Start of Assistant B's response] {R2} [The End of Assistant B's response]
[System]
We would like to request your feedback on the performance of two AI assistants in response to the user question displayed above.
Please rate the helpfulness, relevance, accuracy, level of details of their responses. Each assistant receives an overall score on a scale of 1 to 10, where a higher score indicates better overall performance. Please first output a single line containing only two values indicating the scores for Assistant A and B, respectively. The two scores are separated by a space. In the subsequent line, please provide a comprehensive explanation of your evaluation, avoiding any potential bias and ensuring that the order in which the responses were presented does not affect your judgment.
We would like to request your feedback on the performance of one AI assistants in response to the user question displayed above.
Please rate the helpfulness, relevance, accuracy, level of details of their responses. The assistant receives an overall score on a scale of {min_score} to {max_score} (with a minimum interval of {interval}), where a higher score indicates better overall performance.
Please first output a single line containing only one value indicating the score for Assistant. In the subsequent line, please provide a comprehensive explanation of your evaluation, avoiding any potential bias and ensuring that the order in which the responses were presented does not affect your judgment.

---

Table 8: The score-based evaluation form for single-wise comparison with six slots ({Q}, {R1}, {R2}, {min_score}, {max_score}, {interval} ).

In this section, following the same setting as Zheng et al. (2023), we conduct a preliminary study of single-wise score-based LLM comparison. We use the template shown in Table 8 to generate the input for LLM evaluators. For each question, we generate three sets of value ranges, setting min_score to 0, max_score to 1, 10, and 100, and interval to 0.1, 1, and 10, respectively. In theory, if the single-wise answer is steady and robust, the score should scale accordingly to the value ranges. For example, if the score is 0.7 when the max_score is 1, the score should be 7 when the max_score is 10, and 70 when max_score is 100.

The LLM evaluators are asked to score each answer independently. We use the answers from "llama-13b" as the input for LLM evaluators, and choose "GPT-3.5" as the LLM evaluator. Among a total of 80 test cases, we find that the single-wise comparison does not remain consistent for any of them. Therefore, we conclude that the absolute scores of single-wise comparison do not strictly adhere to a linear mapping relationship across different scales, potentially undermining their significance. It is worth noting that although single-wise comparison has been used in prior research by Chiang & yi Lee (2023); Liu et al. (2023); Zheng et al. (2023) to evaluate open-ended questions. It does not involve comparing two responses together, thereby eliminating any position bias. As a result, our paper primarily focuses on the position bias in pairwise comparison.

# C  PORTIA'S PIPELINE

This section explains the full pipeline of utilizing PORTIA for LLM evaluation. As depicted in Figure 4, typically there are four key steps: (1) **Data preparation**, (2) **Comparison method selection**, (3) **Evaluation**, and (4) **Answer extract**.

In the first step, we prepare the data for evaluation, which includes the questions and corresponding answers from two different LLMs to be compared. If PORTIA is not implemented, we next choose the comparison method and formulate the input prompt, which has a great impact on the evaluation results, as we discussed in Section 4.2. The selected LLM evaluator is then queried with this prompt to obtain judgments. Note that the raw evaluation results require additional processing for two reasons: (1) the output format may differ from the desired final verdicts and (2) the LLM evaluators may deviate from expected responses. For example, the LLM evaluator may fail to return the likert

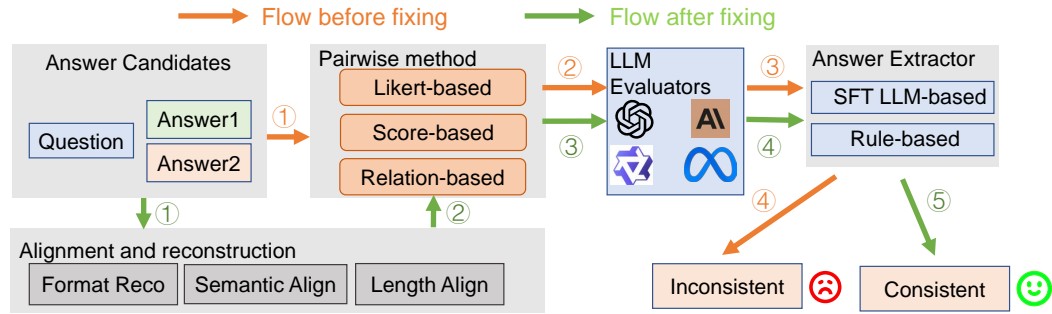

Figure 4: This is the overview of using PORTIA for LLM evaluation. "Reco" and "SFT" are short for "recognition" and "supervised fine-tuning", respectively.

score for some questions but instead return the final verdict directly. Therefore, we design an answer extractor to extract the final verdict from the evaluation results. Specifically, we adopt a hybrid method to extract the final verdict, which first tries to extract with a rule-based system, and if it fails, then it tries with a supervised fine-tuning Chatglm2 (Zeng et al., 2022) model.

The PORTIA-enhanced evaluation would necessitate an additional step of alignment and reconstruction, which constitutes the core of our framework. As elucidated in the main text, this procedure is vital for assessing the LLM answers' quality with less position bias.

## D   LLM DETAILS

In this section, we provide more details about the LLM evaluators and answers used in our experiments.

**LLM Evaluators.** We evaluate PORTIA using six distinct LLMs as evaluators:

- **GPT-4** (OpenAI, 2023) is a large multimodal model capable of processing image and text inputs to generate text outputs. GPT-4 demonstrates human-level aptitude on various professional and academic benchmarks. We utilize the 8K context length "gpt-4-0613" configuration by default.

- **GPT-3.5**  is a 175B parameter model from OpenAI offered in 4K and 16K context length versions. Our experiments use the 4K context "gpt-3.5-turbo-0301" model as default.

- **Claude2** (cla) is the latest large language model released by Anthropic. It supports at most 100k tokens as input. We leverage the default Claude2 API in our tests.

- **Llama2** (Touvron et al., 2023), an open-source series of LLMs from Meta AI ranging from 7B to 70B parameters, is trained on 2 trillion tokens and doubles Llama1's context length. Its fine-tuned iterations utilize over 1 million human annotations. We evaluate both 7B and 13B Llama2 chat models.

- **Qwen** (qwe) is a partially open-sourced LLM model released by Alibaba.  We use the default API service provided by Alibaba cloud in our experiments.

- **Chatglm2** (Zeng et al., 2022) is the second-generation version of the open-source bilingual chat model ChatGLM-6B. We use the offered 6B version in our experiments.

**LLM answers.** As mentioned in Section 4.1, we consider eight answer combinations from different LLMs, specifically, the pairs are: "gpt-3.5-turbo" versus "claude-v1", "llama-13b" versus "vicuna-13b", "alpaca-13b" versus "vicuna-13b", "gpt-3.5-turbo" versus "gpt-4", "gpt-4" versus "claude-v1", "vicuna-13b" versus "vicuna-7b", "vicuna-7b" versus "alpaca-13b", and "gpt-4" versus "vicuna-13b". Note that the answers are generated by the LLMs without any post-processing, and we reuse these answers from previous work (Zheng et al., 2023).

# E  LM METRIC

In this section, we first introduce the LM metric used in our experiments. Then we conduct a controlled experiment to find the optimal number of splits $k$ across different metrics in terms of performance and efficiency.

**LM Metric.** We use the Sentence-BERT (Reimers & Gurevych, 2019) to measure the similarity between pairs. Sentence-BERT is a modification of the pretrained BERT (Devlin et al., 2019) network that uses siamese and triplet network structures to derive semantically meaningful sentence embeddings that can be compared using cosine-similarity. This is efficient while maintaining the accuracy of BERT.

**Efficiency Evaluation.** We use the same setup as in Section 4.1 to conduct the experiment. According to the theoretical analysis in Section 4.3, we set $k \in \{1, 2, 3, 4\}$ and evaluate their efficiency, the results are shown in Table 9. Note that $k$ is the number of segments after splitting, thus $k = 1$ means no splitting would be performed, which leads to 0 in terms of execution time. In short, it can be interpreted from the table that the execution time grows exponentially with the increasing $k$.

|  | $k = 1$ | $k = 2$ | $k = 3$ | $k = 4$ |
|---|---|---|---|---|
| Token-overlap | 0 | 0.31 | 3.71 | 33.12 |
| Bert-model | 0 | 2.37 | 21.3 | 295.10 |

Table 9: Average execution time per input of different metrics with different $k$.

|  | $k = 1$ | $k = 2$ | $k = 3$ | $k = 4$ |
|---|---|---|---|---|
| Token-overlap | - | 53.3 | 66.7 | 73.3 |
| Bert-model | - | 55.9 | 66.7 | 66.7 |

Table 10: Fixed coverage rates of different metrics with different $k$.

**Performance Evaluation.** Following the experimental setup described above, we set $k \in \{1, 2, 3, 4\}$ and evaluate their performance. To clarify, we use the answers from the LLM "gpt-3.5-turbo" and "claude-v1" in our experiments (under the same conditions outlined in Section 4.4), where in total of 80 questions are fed to GPT-3.5 for evaluation. The results are shown in Table 10, where we can see that with the increasing $k$, the fixed coverage rates of both metrics are increasing, and when $k = 3$, the fixed coverage rate of both metrics is the same, which is 66.7%. However, further increasing $k$ results in limited additional gains in coverage. Considering the execution time which grows exponentially with the increasing $k$, we choose $k = 3$ with token-overlap as the default setting in our experiments.

# F  ALGORITHM ILLUSTRATION

To elucidate the operational details of the proposed splitting algorithm, we provide a schematic depiction in Figure 5. Given two LLM-generated answers, the algorithm first identifies all candidate split positions coinciding with sentence boundaries in each answer. It then performs length alignment by initially dividing each answer equally into $k$ partitions and recording the corresponding split positions. Next, for each answer, the split position closest to the recorded locations is selected from the candidate positions. The answers are partitioned into $k$ segments at these chosen split positions. The resulting segments are fed as inputs to the LLM evaluator to obtain the respective judgments.

In cases where inconsistent judgments persist, the algorithm proceeds with semantic alignment to further divide each answer into $k$ parts. Specifically, an iterative search is conducted for optimal split positions that maximize the cumulative semantic similarity between the corresponding segments from the two answers. This traversal terminates when the complete set of potential split positions has been evaluated. Through this process based on both length and semantic alignment, the algorithm is able to decompose the LLM answers into aligned parts for more consistent and reliable evaluation.

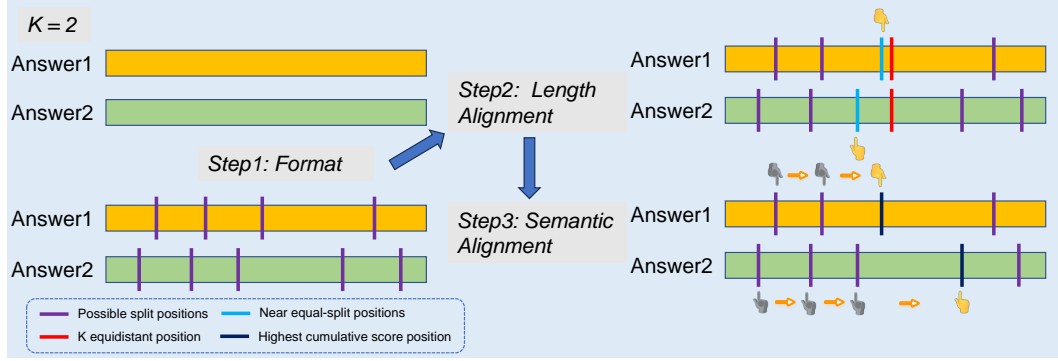

Figure 5: Schematic illustration of the proposed splitting algorithm, depicting its operation when configured with $k = 2$ (i.e., division into two parts).

---

**Algorithm 2:** Details of Step 3: Semantic Alignment ($k = 2$)

```
/* Step3:  semantic alignment.                                         */
```
1   $s_{max} = 0, r_1^{bestparts} = [], r_2^{bestparts} = []$
2   **for** $i$ in $range(len(r_1^{positions}))$ **do**
3     **for** $j$ in $range(len(r_2^{positions}))$ **do**
4       $pos_1 = r_1^{positions}[i], pos_2 = r_2^{positions}[j]$
5       $r_1^{parts}[0] = r_1[: pos_1], r_2^{parts}[0] = r_2[: pos_2]$
6       $r_1^{parts}[1] = r_1[pos_1 :], r_2^{parts}[1] = r_2[pos_2 :]$
7       $s_{cum} = \sum_{i=1}^{2} similarity(r_1^{parts}[i], r_2^{parts}[i])$
```
           /* Update max similarity score, keep best split positions.     */
```
8       **if** $s_{cum} > s_{max}$ **then**
9         $s_{max} = s_{cum}, r_1^{bestparts} = r_1^{parts}, r_2^{bestparts} = r_2^{parts}$
10       **end**
11     **end**
12 **end**

---

**Algorithm 3:** Details of Step 3: Semantic Alignment ($k = 3$)

```
/* Step3:  semantic alignment.                                         */
```
1   $s_{max} = 0, r_1^{bestparts} = [], r_2^{bestparts} = []$
2   **for** $i_1$ in $range(len(r_1^{positions}))$ **do**
3     **for** $i_2$ in $range(i_1 + 1, len(r_1^{positions}))$ **do**
4       **for** $j_1$ in $range(len(r_2^{positions}))$ **do**
5         **for** $j_2$ in $range(j_1 + 1, len(r_2^{positions}))$ **do**
6           $pos_{11} = r_1^{positions}[i_1], pos_{21} = r_2^{positions}[j_1]$
7           $pos_{12} = r_1^{positions}[i_2], pos_{22} = r_2^{positions}[j_2]$
8           $r_1^{parts}[0] = r_1[: pos_{11}], r_2^{parts}[0] = r_2[: pos_{21}]$
9           $r_1^{parts}[1] = r_1[pos_{11} : pos_{12}], r_2^{parts}[1] = r_2[pos_{21} : pos_{22}]$
10          $r_1^{parts}[2] = r_1[pos_{12} :], r_2^{parts}[2] = r_2[pos_{22} :]$
11          $s_{cum} = \sum_{i=1}^{3} similarity(r_1^{parts}[i], r_2^{parts}[i])$
```
                   /* Update max similarity score, keep best split positions.
                   */
```
12          **if** $s_{cum} > s_{max}$ **then**
13            $s_{max} = s_{cum}, r_1^{bestparts} = r_1^{parts}, r_2^{bestparts} = r_2^{parts}$
14          **end**
15         **end**
16       **end**
17     **end**
18 **end**

## G  ON LLAMA2

Although LLMs can be effective evaluators, they are not infallible. In some cases, LLM evaluators may struggle to provide a meaningful verdict. We find that these models would become too conservative to make a verdict, no matter how the answers are split and aligned. Below is an example of the Llama2 models (Touvron et al., 2023) that are not well-served as evaluators.

---

Assistant A: Alpaca-13b, Assistant B: Vicuna-13b
[Question] {How can I improve my time management skills?}
[The Start of Assistant A's response]
Improving time management skills involves setting priorities, breaking tasks into smaller chunks, delegating tasks, avoiding multitasking, and taking regular breaks. Additionally, it is important to have a positive attitude and be realistic with goals. Making a to-do list, tracking time, and using technology to automate mundane tasks can also help improve time management skills.
 [The End of Assistant A's response]
[The Start of Assistant B's response]
Improving your time management skills can help you to be more productive, focused, and less stressed. Here are some tips to help you improve your time management skills: 1. Set clear goals: Identify what you want to achieve and set specific, measurable, and realistic goals. Prioritize your tasks and focus on the most important ones first. 2. Use a calendar or planner: Use a tool to organize your time and schedule your tasks. This can help you to stay on track and make the most of your time. 3. Break down tasks: Break down larger tasks into smaller, more manageable steps. This can help you to focus on one step at a time and make steady progress. 4. Eliminate distractions: Identify and eliminate distractions that may be preventing you from being productive. This can include turning off your phone, closing unnecessary tabs on your computer, or finding a quiet place to work. 5. Practice self-discipline: Use self-discipline to stay focused and avoid procrastination. Set a timer for each task and challenge yourself to complete it before the timer goes off. 6. Take breaks: Take regular breaks to rest and recharge. This can help you to stay focused and avoid burnout. 7. Review your progress: Regularly review your progress and adjust your strategy as needed. This can help you to stay on track and make improvements over time.
 [The End of Assistant B's response]

Evaluator: {Llama2-7b-chat}, Form: {score-based}, Assessment: {\n\n\n\n\n\n\n\n }
Evaluator: {Llama2-7b-chat}, Form: {likert-based}, Assessment: {""}
Evaluator: {Llama2-13b-chat}, Form: {score-based}, Assessment: {Please proceed with your evaluation.}
Evaluator: {Llama2-13b-chat}, Form: {likert-based}, Assessment: {\n\nPlease provide your feedback.}

---

Table 11: The assessments of Llama2-7b-chat and Llama2-13b-chat on two comparison forms.

