# OpenReview forum: "Split and Merge: Aligning Position Biases in Large Language Model based Evaluators"
_ICLR.cc/2024/Conference — Submitted to ICLR 2024_

### Official Review · Reviewer_cPgm · 2023-10-20

**Soundness:** 3 good
**Presentation:** 3 good
**Contribution:** 3 good
**Rating:** 5
**Confidence:** 4

**Summary:**

This paper introduces a novel evaluation system called Portia based on large language models (LLMs). Portia is designed to mitigate positional bias in pairwise comparison with LLMs. Instead of presenting candidate answers sequentially, Portia splits each answer into segments, aligns these segments on 2 sides based on their similarities, and then combines segment pairs to construct prompts for LLMs. Portia exhibits a notable enhancement in answer agreement compared to the baseline method, effectively rectifying most of the inconsistencies found in the baseline models. Furthermore, the evaluations generated by Portia demonstrate a closer alignment with both human evaluations and outputs from GPT-4. Lastly, the authors conduct a cost analysis of Portia, identifying an optimal segment number that strikes a balance between cost and effectiveness.

**Strengths:**

1. The paper is well written with a clear description of the method and appropriate discussions.
2. As outlined in the summary section, Portia can be practically useful as it improves the consistency of LLMs for pairwise comparison and improves the assessment quality, especially for smaller LLMs.
3. The experiment design is reasonable, covering many popular LLMs and 3 comparison forms.
4. The paper includes extensive appendices and supplementary materials, which makes the paper easy to reproduce and more understandable.

**Weaknesses:**

1. It is in doubt about the semantic meaningfulness of alignment. The method assumes that answers can be deconstructed, allowing for rough alignment on a segment-by-segment basis. However, this assumption may not universally hold for open-ended answers. It is common for two models to generate entirely distinct yet equally valid responses. Additionally, they may produce similar content with variations in the order, as exemplified in the question section. In many scenarios, alignment may not yield meaningful results.
2. Limited applicability: The proposed system, i.e. the pipeline of splitting, alignment, and merging, is a complicated rule-based system, which can be brittle. The authors only test their system on MT benchmark, so it is interesting to see if a system that is fine-tuned toward MT benchmark can be applicable to other forms of assessment tasks.

**Questions:**

1. The paper mentions that the split should preserve the answer order, then it may cause an issue of alignment when the answer is order insensitive. What if 2 models generate segments of texts with different orders, but both are fine? E.g. the question is "What should you do in the morning after waking up?". Model A generates "brush teeth and wash face" and model B generates "wash face and brush teeth", then an oder-invariant alignment will not be able to find an optimal alignment.
2. How does Portia work for short sequences, which cannot be split into shorter segments?

---

> ### Author Response · Authors · 2023-11-16
> **Response to Reviewer cPgm**
>
> ### Response to Reviewer cPgm
>
> #### Q1
> > It is in doubt about the semantic meaningfulness of alignment. The method... In many scenarios, alignment may not yield meaningful results.
>
> > The paper mentions that the split should preserve the answer order, then it may... Then an oder-invariant alignment will not be able to find an optimal alignment.
>
>
> Thank you for your question. As we clarified in our **global response**, our
> framework is designed to accommodate a range of open-ended questions and
> responses generated by various LLMs. While Figure 1 depicts point-to-point
> responses, this represents only a small portion (less than 10%) of the total
> responses, which are predominantly open-ended.
>
> About the order insensitive answer, we clarify that determining whether two
> sentences remain semantically equivalent after sequential exchanges is a
> long-standing and complex problem [5,7], and our work does not attempt to
> address this issue. It is possible that the requirement of our "order
> preservation" is so strict that we may miss some cases where the order is not
> important. However, we argue that "order preservation" is a reasonable
> requirement for most cases, as it is hard to determine whether the order is
> important or not.
>
> Moreover, the goal of this work is to address the position bias issue in LLM
> evaluators, which have been proven to be capable of providing high-quality
> evaluations compared to human evaluators [1,2]. To achieve this goal, our
> framework tries to identify a segmentation choice with balanced length (length
> alignment) or better semantic overlap (semantic alignment). Both alignment
> methods serve to enhance the consistency of evaluation results, as outlined in
> Section 4.5.
>
>
>
> #### Q2
>
> > How does Portia work for short sequences, which cannot be split into shorter segments?
>
> Thank you for pointing this out. Please first refer to our **global response**
> for the discussion of the responses' length.
>
> Moreover, we clarify that we do not have a strict requirement for the length of
> the responses. As long as the response has content that can be segmented (i.e., >k), our framework is applicable. During our experiments, we found that it is
> uncommon for responses to be too short to be segmented, and in such cases, our
> framework would not make any enhancements. This is because, as discussed in our
> global response, the position bias is typically not present in extremely short
> answers, and thus no adjustments are necessary.
>
>
> #### Q3
>
> > Limited applicability: The proposed system, i.e. the pipeline of splitting, alignment... so it is interesting to see if a system that is fine-tuned toward MT benchmark can be applicable to other forms of assessment tasks.
>
>
>
> Thank you for your insightful suggestion. As we stated in the **global
> response**, the benchmark we select includes a diverse range of open-ended
> questions from eight primary categories, such as writing, reasoning, and
> roleplay. The eight distinct LLMs that are considered in this work possess
> varying capabilities and preferences, resulting in diverse responses. Furthermore, this benchmark has been widely used in
> previous research [3,4], indicating its applicability to a variety of assessment
> tasks.
>
> Moreover, we acknowledge that it would be interesting to investigate whether
> position bias exists in other benchmarks such as alpaca_eval [6]. This benchmark
> contains shorter answers, with 53% of the responses being less than 412
> characters. As shown in the **global response**, it is difficult for LLM
> evaluators to exhibit position bias when the answers are shorter. Therefore,
> exploring position bias in other benchmarks with shorter answers could provide
> valuable insights, and we leave this as a potential direction for future
> research.
>
>
> [1] Cheng-Han Chiang and Hung-yi Lee. 2023. Can Large Language Models Be an Alternative to Human Evaluations?. In Proceedings of the 61st Annual Meeting of the Association for Computational Linguistics (Volume 1: Long Papers), pages 15607–15631, Toronto, Canada. Association for Computational Linguistics.
>
> [2] Liu, Yang, et al. "Gpteval: Nlg evaluation using gpt-4 with better human alignment." arXiv preprint arXiv:2303.16634 (2023).
>
> [3] Zheng, Lianmin, et al. "Judging LLM-as-a-judge with MT-Bench and Chatbot Arena." arXiv preprint arXiv:2306.05685 (2023).
>
> [4] Wang, Peiyi, et al. "Large language models are not fair evaluators." arXiv preprint arXiv:2305.17926 (2023).
>
> [5] Schwartz, Daniel G. "Axioms for a theory of semantic equivalence." Fuzzy Sets and Systems 21.3 (1987): 319-349.
>
> [6] https://huggingface.co/datasets/tatsu-lab/alpaca_eval
>
> [7] Finch, Andrew, Young-Sook Hwang, and Eiichiro Sumita. "Using machine translation evaluation techniques to determine sentence-level semantic equivalence." Proceedings of the third international workshop on paraphrasing (IWP2005). 2005.

---

> ### Comment · Reviewer_cPgm · 2023-11-21
>
> Thanks for your response. I agree with other reviewers and think the major issue of this paper is still the generalizability. The experiments in the global response do not alleviate my concerns: It shows that the variable lengths do not affect the consistency, but the point of "split and merge" seems groundless in this scenario. Moreover, it does not explain how Portia works for semantically different responses.

---

> > ### Author Response · Authors · 2023-11-22
> > **Response to Reviewer cPgm**
> >
> > ### Response to Reviewer cPgm
> >
> >
> > > Thanks for your response. I agree with other reviewers and think the major issue of this paper is still the generalizability. The experiments in the global response do not alleviate my concerns: It shows that the variable lengths do not affect the consistency, but the point of "split and merge" seems groundless in this scenario. Moreover, it does not explain how Portia works for semantically different responses.
> >
> > Thank you for spending the time to read our rebuttal. We would like to first clarify a misunderstanding here.
> > The objective of these experiments is not to demonstrate that variable lengths do not impact consistency, but rather to show that *there is a generally positive correlation between answer length and inconsistency rate*.
> >
> > Furthermore, the global response experiments also highlight two key findings: (1) The responses in the selected benchmark are diverse. (2) The proposed method effectively addresses the position bias issue associated with varying responses. This is demonstrated even in the extreme scenario where the answer is forced to be extremely short, which is unlikely to occur in normal usage of LLMs.
> > Specifically, the responses in "gpt-3.5-short" and "gpt-3.5" represent the situation that two responses roughly have similar content but different lengths, while the responses in "gpt-3.5-short" and "claude-v1" represent the situation that two responses have different content and length. No position bias exists in any situation.
> >
> > Through these experiments, we hope to demonstrate the versatility of Portia as a general framework capable of accommodating a wide range of responses in terms of experimental performance. In this response below, we would like to further present the algorithmic principles behind Portia, and explain why Portia is a general framework that works for semantically different responses.
> >
> >
> > **Algorithmic principles**
> >
> > In Section 3, we outlined our design intuition for Portia, which involves splitting candidate answers into shorter segments to address the position bias that LLM evaluators face when dealing with lengthy and intricate answers.
> > The primary reason for merging the segments back together is to ensure that the original content of the answer is preserved, which is critical for accurate evaluation.
> > Algorithm 1 illustrates the design of Portia, which comprises length alignment and segment alignment, and both of these techniques are effective in mitigating the position bias issue, as discussed in Section 4.5.
> >
> >
> > Notably, the semantic differences between responses did not influence the algorithmic principles underlying the design of Portia. LLM evaluators have demonstrated their proficiency in evaluating responses, regardless of their semantic dissimilarity.
> > The primary challenge addressed by Portia is the position bias issue that LLM evaluators encounter.
> > Therefore, we *do not attribute the ability to evaluate semantically different responses to Portia, but rather to the LLM evaluators themselves.*
> >
> >
> >
> > **Summary**
> >
> > Overall, the generalizability of Portia is evident in two key aspects.
> > First, the algorithm's design is highly generalizable, with relaxed requirements in terms of the length and content of input responses, making it adaptable to a wide range of situations.
> > Additionally, experimental results demonstrate that Portia is able to maintain a high enhancement rate across a broad spectrum of responses. This demonstrates the algorithm's ability to generalize effectively, making it a valuable tool for a variety of applications.

---

### Official Review · Reviewer_SWLn · 2023-10-30

**Soundness:** 1 poor
**Presentation:** 2 fair
**Contribution:** 2 fair
**Rating:** 3
**Confidence:** 5

**Summary:**

This paper studies the problem of using LLMs as evaluators. They focus on pair-wise comparison, where the LLM is given a pair of responses to a question and it needs to select which one is better. They aim to mitigate the position bias of LLM evaluation, which is the case that an LLM evaluation result is different by swapping the two responses.

This paper proposes a pipeline, Portia, to mitigate the position bias in pair-wise LLM evaluation. This is done by the following process: Given two responses, they split each response into an equal number of fragments with roughly the same size, and they pair the fragments from the two responses together and let the LLM evaluate.

The results show that Portia can largely mitigate the positional bias of 6 LLMs including Claude-2, Llama2, and ChatGPT. They also show that Portia is a method that can close the gap between GPT-4 and ChatGPT while costing less resources (money and time)

**Strengths:**

- Positional bias in LLM evaluation is an important problem that needs to be solved.
- The proposed method can successfully reduce the positional bias of all the six LLMs they tested on.

**Weaknesses:**

- **The method is not clearly described, making it hard for me to properly evaluate the soundness and contribution of the paper**. Specifically, the semantic alignment part is incomprehensible. The text on page 5 does not provide too much information, and Algorithm 1 also does not clearly explain what it is doing. The functions $partition$ and $n_s$ are not explained and are hard to understand. The appendix also does not provide any further elaborations. So I also cannot evaluate the correctness of the analysis on page 7. Considering that this is a core part of the proposed method, **I cannot assess this paper if this issue is not resolved.** I am willing to adjust my review based on the author's response.

- The models used in this paper are not very well explained. I cannot see which Llama-2 model is used, the Llama-2 model or the Llama-2-chat model.

- The experiment setting in Appendix B is not clear. The paper seems to require that the scores in different scales need to have a linear mapping relationship. I do not agree with this. As long as we can use the scores derived from LLMs ratings to obtain meaningful comparison. the scores are meaningful. So a more reasonable comparison may be calculating the correlation coefficient as prior works that use single-wise comparison. Refer to [1, 2] for more experiment details.

- Missing references: [1] and [2] are the earliest two works that use LLMs as evaluators and should be included in the related works.


[1] Cheng-Han Chiang and Hung-yi Lee. 2023. Can Large Language Models Be an Alternative to Human Evaluations?. In Proceedings of the 61st Annual Meeting of the Association for Computational Linguistics (Volume 1: Long Papers), pages 15607–15631, Toronto, Canada. Association for Computational Linguistics.

[2] Liu, Yang, et al. "Gpteval: Nlg evaluation using gpt-4 with better human alignment." arXiv preprint arXiv:2303.16634 (2023).

**Questions:**

Q1. What happens if the responses to be rated are very short, or what if the two responses that need to be rated have a very different length?

I may have further questions after the authors respond to the weaknesses part.

---

> ### Author Response · Authors · 2023-11-16
> **Response to Reviewer SWLn**
>
> ### Response to Reviewer SWLn
>
> #### Q1
>
> > The method is not clearly described, making it hard for me to properly evaluate the soundness and contribution of the paper. ... I am willing to adjust my review based on the author's response.
>
>
> Thank you for pointing this out. We appreciate your valuable feedback and take
> steps to address the concerns you raised. Specifically, we make revisions to the
> description of the semantic alignment part in Section 3.2 to provide a more
> comprehensive and understandable explanation. We clarify that $n_s$ represents
> the index number of the current segmentation, and its maximum value is
> equivalent to the total number of computation operations $Cal$ as discussed in
> Section 4.3. Additionally, we emphasize that the depth of loop traversal is
> dependent on the chosen value of $k$, and hence, we have used $n_s$ to denote
> the extent of traversal.
>
> To further enhance the clarity of our framework, we include two detailed
> algorithm descriptions of the function $partition$ in Appendix F. These
> descriptions provide a step-by-step process of semantic alignment when $k$ is
> equal to 2 and 3, respectively. We believe that these revisions will provide a
> more comprehensive and lucid explanation of our framework. [Please refer to our
> revised paper for all the details. We have marked major changes in red.]
>
>
> #### Q2
> > The models used in this paper are not very well explained. I cannot see which Llama-2 model is used, the Llama-2 model or the Llama-2-chat model.
>
>
> Thank you for your question. We apologize for any confusion caused by the lack
> of clarity regarding the specific Llama-2 model used in our paper. We would like
> to clarify that the Llama-2-chat models are used in our experiments, and
> necessary revisions are made to Appendix G of our manuscript to reflect this.
>
>
> #### Q3
> > The experiment setting in Appendix B is not clear. The paper seems to require that the scores in different scales need to have a linear mapping relationship... Refer to [1, 2] for more experiment details.
>
>
> Thank you for your insightful feedback and suggestions. We would like to clarify
> that the primary focus of our paper is on pairwise comparison, as it is
> particularly relevant to our study's scope. Since single-wise comparison does
> not require comparing two responses together, there is no position bias. We
> acknowledge that single-wise comparison is a valid approach for evaluating
> open-ended questions, and it has been employed in previous research, including
> the works the reviewer mentioned [1,2], as well as other recent studies [3,4,5].
>
> We apologize for any confusion our description of the experiment setting in
> Appendix B may have caused. Our intention is to emphasize that the absolute
> scores do not strictly follow a linear mapping relationship in different scales,
> which may weaken their significance. We understand that our statement regarding
> the inconsistency of single-wise comparison may have been overstated in Section
> 2.
>
> In our revised manuscript, we provide a more balanced discussion of both
> single-wise and pairwise comparison methods in Appendix B, acknowledging the
> merits of each approach. We emphasize the primary focus of our paper and ensure
> that our descriptions are clear and accurate, reflecting the limitations and
> scope of our study.
>
>
> #### Q4
> > Missing references: [1] and [2] are the earliest two works that use LLMs as evaluators and should be included in the related works.
>
> Thank you for bringing to our attention the missing references [1] and [2] which
> are the earliest works that use LLMs as evaluators. We appreciate your
> suggestion and include these references in the related works section of our
> revised manuscript to provide a comprehensive overview of the relevant
> literature.
>
>
>
> #### Q5
> > What happens if the responses to be rated are very short, or what if the two responses that need to be rated have a very different length?
>
> Thank you for your question. Please refer to our **global response**.
>
>
>
> [1] Cheng-Han Chiang and Hung-yi Lee. 2023. Can Large Language Models Be an Alternative to Human Evaluations?. In Proceedings of the 61st Annual Meeting of the Association for Computational Linguistics (Volume 1: Long Papers), pages 15607–15631, Toronto, Canada. Association for Computational Linguistics.
>
> [2] Liu, Yang, et al. "Gpteval: Nlg evaluation using gpt-4 with better human alignment." arXiv preprint arXiv:2303.16634 (2023).
>
> [3] Zheng, Lianmin, et al. "Judging LLM-as-a-judge with MT-Bench and Chatbot Arena." arXiv preprint arXiv:2306.05685 (2023).
>
> [4] Wang, Peiyi, et al. "Large language models are not fair evaluators." arXiv preprint arXiv:2305.17926 (2023).
>
> [5] Zeng, Zhiyuan, et al. "Evaluating large language models at evaluating instruction following." arXiv preprint arXiv:2310.07641 (2023).

---

> > ### Comment · Reviewer_SWLn · 2023-11-21
> >
> > Thank you for your response. The algorithm is more readable now.
> >
> > However, after reading the responses from the authors and the reviewers of fellow reviewers, I will stick to my original evaluation.
> > I consider the method to be limited to cases when the two responses differ in length significantly **and have different content**. In the supplementary experiment, the authors claimed that LLM-as-a-judge does not suffer from position bias when the two responses differ in length. However, this is a very limited evaluation since the two responses used in the experiment essentially have the same content. Whether position bias exists when the two responses differ in content and length is unclear. Moreover, the supplementary experiment also does not solve the problem raised by the reviewers: **Can the proposed method deal with the case when two responses differ in length significantly?**

---

> > > ### Author Response · Authors · 2023-11-21
> > > **Response to Reviewer SWLn**
> > >
> > > ## Response to Reviewer SWLn
> > >
> > >
> > > > Thank you for your response. The algorithm is more readable now. However,
> > > after reading the responses from the authors and the reviewers of fellow
> > > reviewers, I will stick to my original evaluation ... Moreover, the
> > > supplementary experiment also does not solve the problem raised by the
> > > reviewers: Can the proposed method deal with the case when two responses differ
> > > in length significantly?
> > >
> > > Thank you for spending the time to read our rebuttal.
> > > In this response, we first conduct further experiments on the gap in response length, and then clarify the misunderstanding in the global response. Finally, we summarize our response.
> > >
> > > **Additional Experiments on Response Length Gap**
> > >
> > > To further explore the relationship between the gap in length between responses
> > > and fixed coverage rate, we conducted an experiment using the collected judgment
> > > data. For this, "GPT-3.5" was used as the evaluator, analyzing 8 pairs of
> > > responses across three comparison forms.
> > >
> > > The answers are categorized into 5 groups based on their length, with each group
> > > representing a 300-character interval. The results are presented below, with
> > > frequencies below 3% of the total being disregarded.
> > >
> > >
> > > |# Char for gap in length|0-300|300-600|600-900|900-1200|1200-1500|
> > > |---|---|---|---|---|---|
> > > |% Fixed coverage|50.82|48.41|63.3|62.67|69.77|
> > > |% Frequency|0.37|0.24|0.17|0.11|0.08|
> > >
> > >
> > > The table shows that the proposed method is consistently effective in addressing
> > > the position bias issue. Furthermore, our analysis reveals a generally positive
> > > correlation between the gap in response length and the fixed coverage rate. When
> > > considering the inconsistent rates observed on "gpt-3.5-short", where responses
> > > with significantly large length gaps did not exhibit the position bias issue, it
> > > can be concluded that *our proposed framework is effective in handling responses
> > > of varying lengths.*
> > >
> > >
> > > **Clarification for Experiments in the Global Response**
> > >
> > >
> > > We wish to clarify a misunderstanding here. Our proposed method can deal
> > > with the case when two responses differ in length significantly. As we explained
> > > in the global response, subsection **Additional Experiments on Response
> > > Length**. The responses in "gpt-3.5-short" and "gpt-3.5" represent the situation
> > > that two responses roughly have similar content but different lengths, while the
> > > responses in "gpt-3.5-short" and "claude-v1" represent the situation that two
> > > responses have different content and length. No position bias exists in any
> > > situation.
> > >
> > > To provide a clearer illustration, consider the following scenario. Suppose the
> > > original question is "As a space colonist on Mars, describe your daily life and
> > > the challenges you face living on another planet." The response generated by
> > > "gpt-3.5" is "As a space colonist on Mars, my daily life would be vastly
> > > different from life on Earth. Some of the main challenges I would face living on
> > > another planet are: ... and our work would contribute to a better understanding
> > > of our universe and our place in it." (1768 characters), while the response
> > > produced by "claude-v1" is "Here is what my daily life as a space colonist on
> > > Mars might look like ... and a pioneering spirit to overcome the many challenges
> > > of establishing a long-term human presence on another world." (1980 characters).
> > >
> > > After shortening, we get "gpt-3.5-short" with a length of "Living on Mars
> > > presents challenges such as limited resources, communication delays, extreme
> > > environment, and ... rely on pre-planned communication" (215 characters). We
> > > then conducted an evaluation study using GPT-3.5 and GPT-4 as evaluators to
> > > compare "gpt-3.5-short" with "gpt-3.5" and "claude-v1" in exchanged orders to
> > > assess consistency. The results of the study indicate that no position bias
> > > exists in either situation.
> > >
> > >
> > >
> > > **Summary**
> > >
> > > Overall, we wish to note that in general, position bias is unlikely to exist when two responses differ in length, regardless of whether they share the same
> > > content or not. This has been supported by additional experiments conducted. In
> > > cases where the responses share similar content, our additional experiments have
> > > demonstrated that position bias is unlikely to occur. Moreover, when responses differ in content (e.g., from different LLMs), LLM evaluators have been shown to have a preference for longer responses [1], and therefore, it is less probable for position bias to occur in such cases.
> > >
> > > In summary, we believe that the proposed method can deal with the case when two
> > > responses differ in length significantly, regardless of whether they share the
> > > same content or not.
> > >
> > > [1] Wang, Yizhong, et al. "How Far Can Camels Go? Exploring the State of Instruction Tuning on Open Resources." arXiv preprint arXiv:2306.04751 (2023).

---

> > > > ### Comment · Reviewer_SWLn · 2023-11-22
> > > >
> > > > Thank you for your clarification. I missed the part on Claude.
> > > >
> > > > However, I still believe the evaluation of length difference is limited.
> > > > In the experiments shown in the rebuttal, the questions are those when longer responses would be generally preferred. So, it should be expected that the LLM will prefer longer responses, disregarding their positions. However, I am more interested in those cases when longer outputs are not always better (for example, two different outputs from different prompting methods, with one longer and one shorter; perhaps even the final answer is different). Or, for example, for a math question, with one short response with only a single sentence that includes the correct number, and a long step-by-step response that ends up with a wrong answer. Can the proposed method work in this case?

---

> > > > > ### Author Response · Authors · 2023-11-22
> > > > > **Response3 to Reviewer SWLn**
> > > > >
> > > > > ## Response to Reviewer SWLn
> > > > >
> > > > >
> > > > > > Thank you for your clarification. I missed the part on Claude.
> > > > > However, I still believe the evaluation of length difference is limited.
> > > > > In the experiments shown in the rebuttal, the questions are those when longer responses would be generally preferred. So, it should be expected that the LLM will prefer longer responses, disregarding their positions. However, I am more interested in those cases when longer outputs are not always better (for example, two different outputs from different prompting methods, with one longer and one shorter; perhaps even the final answer is different). Or, for example, for a math question, with one short response with only a single sentence that includes the correct number, and a long step-by-step response that ends up with a wrong answer. Can the proposed method work in this case?
> > > > >
> > > > >
> > > > >
> > > > > Thank you for your insightful response. To address your concerns, it is important to note that the preference for longer responses is not exclusive to the questions we selected, but is a general phenomenon observed in other datasets as well. In Figure 2 of [1], the authors demonstrate that GPT-4 tends to prefer answers with more unique tokens than those provided by humans. This verbosity bias, as introduced in Section 5, is beyond the scope of our study.
> > > > >
> > > > > Regarding your specific examples, we acknowledge that the LLM evaluator may not be the most suitable choice for certain cases. In the math question scenario, since the correct number serves as ground truth, an exact match evaluation would be more appropriate. For assessing different prompting methods, one can follow the approach in [2], which proposes a checklist for evaluation. Furthermore, when a highly reliable evaluation is required, human evaluation remains a viable option.
> > > > > It is essential to note that numerous methods exist for evaluating LLM responses, and the LLM evaluator is merely one such option. While it offers a cost-effective solution for a wide range of responses compared to exclusive human evaluation, it is more flexible than exact match evaluation, albeit at a relatively higher cost.
> > > > >
> > > > > In summary, since the LLM evaluator is not a method that achieves unquestionably the finest evaluation results in all scenarios, it is unclear whether our framework should be applied. In this work, we focus on the position bias issue for open-ended questions, and we leave the other bias issues and scenarios for future work.
> > > > >
> > > > > [1] Wang, Yizhong, et al. "How Far Can Camels Go? Exploring the State of Instruction Tuning on Open Resources." arXiv preprint arXiv:2306.04751 (2023).
> > > > >
> > > > > [2] Naihin, Silen, et al. "Testing Language Model Agents Safely in the Wild" arXiv preprint arXiv:2311.10538 (2023).

---

### Official Review · Reviewer_YB2D · 2023-10-30

**Soundness:** 3 good
**Presentation:** 3 good
**Contribution:** 3 good
**Rating:** 6
**Confidence:** 4

**Summary:**

This paper presents a method for improving the consistency of using LLM to make choices between two generated texts. The method includes a split-merge process, where the two generated texts are split, aligned and merged into a single text, so that the position bias is reduced.

The writing and organization of the paper is pretty well.

**Strengths:**

The proposed method is simple.

The experiments demonstrated that the consistency is improved for different LLMs by a considerable margin.

**Weaknesses:**

The proposed method seems to be problematic in the following cases:

For answers/texts that are significantly different in length. Or with almost the same length but significantly different in symantics or content.

If the answers/texts have different orders, changing the order of one of the text may affect the evaluation of the generation quality.

Besides, is it possible that the merging operation makes it harder (longer context, in a comparing way) for the LLMs to understand and evaluate the output?

**Questions:**

See the weaknesses.

---

> ### Author Response · Authors · 2023-11-16
> **Response to Reviewer YB2D**
>
> ### Response to Reviewer YB2D
>
> #### Q1
>
> > For answers/texts that are significantly different in length. Or with almost the same length but significantly different in symantics or content.
>
> Thank you for pointing this out. As stated in our **global response**, our
> framework is designed to accommodate responses that may differ significantly. However, we acknowledge that evaluating responses with
> similar lengths but significantly different semantics or content can
> pose a considerable challenge, even for human evaluators. Previous research has
> demonstrated that LLM evaluators possess the ability to provide high-quality
> evaluations [1,2,3] while facing the position bias issue. Our framework
> addresses this issue by splitting and merging responses, and our experimental
> results show that with our framework's assistance, LLM evaluators can provide
> more consistent evaluation results. Furthermore, our framework has the potential
> to improve the agreement rate between human and LLM evaluators.
>
> #### Q2
>
> > If the answers/texts have different orders, changing the order of one of the text may affect the evaluation of the generation quality.
>
> Thank you for pointing this out. As emphasized in Section 3.1 of our framework,
> the preservation of the original sequence of information presented in the
> answers is a crucial consideration in our design. Instead of altering the
> content of the generated responses, our framework adds split boundary prompts
> (as exemplified in Table A.2) between each segment of the merged response. This
> approach allows LLM evaluators to receive the same information as the
> original responses for evaluation purposes.
>
>
> #### Q3
>
> > Besides, is it possible that the merging operation makes it harder (longer context, in a comparing way) for the LLMs to understand and evaluate the output?
>
> Thank you for your question. In Section 2 of our paper, we introduce the concept
> of pairwise comparison as a means of evaluating open-ended questions. This
> approach involves presenting two responses side-by-side for evaluators to
> compare, providing a longer context than the single-wise comparison. Pairwise
> comparison is a widely used method in recent research [1,2,3] due to its ability
> to generate high-quality evaluations.
>
> Our framework, which includes split boundary prompts of several tokens in
> addition to the standard pairwise comparison, is designed to further improve the
> evaluation process. We present experimental results in Section 4.4 that
> demonstrate the effectiveness of our framework in enhancing the agreement rate
> between human evaluators and LLM evaluators. These findings suggest that our
> framework does not impede the ability of LLM evaluators to comprehend and
> evaluate the output.
>
> [1] Zheng, Lianmin, et al. "Judging LLM-as-a-judge with MT-Bench and Chatbot Arena." arXiv preprint arXiv:2306.05685 (2023).
>
> [2] Wang, Peiyi, et al. "Large language models are not fair evaluators." arXiv preprint arXiv:2305.17926 (2023).
>
> [3] Zeng, Zhiyuan, et al. "Evaluating large language models at evaluating instruction following." arXiv preprint arXiv:2310.07641 (2023).

---

### Official Review · Reviewer_RHzc · 2023-11-01

**Soundness:** 2 fair
**Presentation:** 3 good
**Contribution:** 2 fair
**Rating:** 6
**Confidence:** 3

**Summary:**

The paper introduces a new method called PORTIA, which is designed to address position bias in large language model-based evaluators. The method uses semantic and length alignment to calibrate position bias in a lightweight and cost-effective manner. Specifically, PORTIA splits the answers into multiple segments, aligns similar content across candidate answers, and then merges them back into a single prompt for evaluation by LLMs. The authors present experiments with six LLMs and demonstrate the effectiveness of PORTIA in enhancing their consistency rates in serving as evaluators.

**Strengths:**

* The paper addresses an interesting and important limitation of LLM-based evaluators, namely position bias, which has been known to affect the consistency and accuracy of evaluations.
* The proposed method PORTIA is simple to execute.

**Weaknesses:**

* The method design is a bit ad-hoc and does not seem to be generalizable enough for evaluating open-ended generation, considering how flexible/free-form the generation results can be. Specifically, the method requires splitting multiple answers being compared into the same amount of segments, and expecting that the answers are somewhat overlapping semantically. However, this may not work well when the generation results are largely different from each other, and this could be very common in open-ended generation. For example, in creative writing tasks, the answers being compared may not have any semantic overlap, and may even differ a lot in their lengths. It seems that the proposed method does not take these cases into account. Also, I'm not sure if splitting a response into multiple segments is really a good idea, since generation evaluation usually has to consider the answers' coherency (potentially spanning over long-range contexts that should be evaluated as a whole).
* The experiment evaluation only focuses on the consistency rates and does not provide evidence of its impact on the accuracy (e.g., whether the calibrated prediction corresponds better to human judgments). Such evaluations are obviously necessary because any naively deterministic method (for example, a system that always prefers the longer answer) will have a consistent rate of 100% but will not be useful in practice.
* I expect to see stronger baselines included (there are several methods aiming at enhancing LLM-as-evaluators, though they may not be completely addressing position bias). I'd encourage the authors to also discuss the new studies that came out after the ICLR submission deadline such as Zeng et al. (this is not a weakness but a suggestion)

Reference:
Zeng et al. “Evaluating Large Language Models at Evaluating Instruction Following.” ArXiv abs/2310.07641

**Questions:**

Please address the weaknesses mentioned above.

---

> ### Author Response · Authors · 2023-11-16
> **Response to Reviewer RHzc**
>
> ### Response to Reviewer RHzc
> #### Q1
>
> > The method design is a bit ad-hoc and does not seem to be generalizable enough for evaluating open-ended generation, ...., since generation evaluation usually has to consider the answers' coherency (potentially spanning over long-range contexts that should be evaluated as a whole).
>
>
> Thank you for pointing this out. In our **global response**, we clarify that our
> framework is applicable to open-ended questions, where the responses can be
> largely different from each other.
>
> Furthermore, we clarify that our framework does not involve splitting a response
> into multiple segments and querying LLM-based evaluators multiple times for
> separate evaluations. This approach may compromise the coherence of the response and incur considerable extra cost. Instead, our framework splits and merges
> the response, and the evaluator is queried only once with the merged response.
>
> To facilitate the evaluators' ability to distinguish between the two responses
> and avoid misidentification, we include split boundary prompts (as exemplified
> in Table A.2) between each segment of the merged response. As a result, the
> LLM-based evaluators receive the same information as the original responses, in
> the same order, and with clear boundaries. This assists the evaluators in
> evaluating the responses as a cohesive whole.
>
>
> #### Q2
>
> > The experiment evaluation only focuses on the consistency rates and does not provide evidence of its impact on the accuracy (e.g., whether the calibrated prediction corresponds better to human judgments). Such evaluations are obviously necessary because any naively deterministic method (for example, a system that always prefers the longer answer) will have a consistent rate of 100% but will not be useful in practice.
>
> Thank you for your question. We would like to clarify that in Section 4.4 of our
> study, we have presented evidence regarding the impact of our framework on
> assessment accuracy. Specifically, Table 3 displays the results of our
> evaluation, demonstrating that our framework is effective in enhancing agreement
> rates between human and LLM evaluators. The table reveals that the average human
> agreement on original LLM assessments increases by 11.25% after using our
> framework. Further analysis can be found in Section 4.4. These quantitative
> results shall provide compelling evidence that our framework successfully
> augments the assessments of all LLM evaluators, resulting in greater accuracy.
>
> #### Q3
>
> > I expect to see stronger baselines included (there are several methods aiming at enhancing LLM-as-evaluators, though they may not be completely addressing position bias). I'd encourage the authors to also discuss the new studies that came out after the ICLR submission deadline such as Zeng et al. (this is not a weakness but a suggestion)
>
> **Stronger baselines**
>
> Thank you for your suggestion. Following your advice, we compare our framework with various baselines,
> including the vanilla method, MEC, BPC, and HITLC. Specifically,
> the VANILLA method asks evaluators to simply output their preference without any explanation, while the prompts used in our baseline are presented in Appendix A.1.
> MEC, BPC, and HITLC are the methods proposed in [2], which require multiple
> turns of querying or human effort.
>
> We present the results in the table below, where the agreement rate between humans and corresponding LLM evaluators for GPT-4 and GPT-3.5 are provided.
>
>
> ||GPT-4|GPT-3.5|Avg Cost|
> |--|--|--|--|
> |VANILLA [1]|52.7|44.4| 1x|
> |Our baseline|60.0|55.0| 1.03x|
> |MEC [2]|60.9|55.6|3.29x |
> |MEC+BPC [2]|62.5|58.7|3.29x |
> |Ours|65.0|63.8|1.68x|
> |Human-in-the-loop (HITLC) [2]|73.8|71.3| 97.3x|
>
> From the table, we can observe that our framework outperforms all methods except
> HITLC, which requires human effort at an extremely high cost. Given that our
> framework is a fully automated method with low cost, we believe that it serves
> as a strong baseline for future research.
>
>
> **New studies**
>
> Thank you for your valuable suggestion. We have considered it and have made the necessary revisions to our manuscript. In particular, we have
> included the findings of [3] that position bias exists in the evaluation of instruction-following models in Section 1.
>
> [1] Zheng, Lianmin, et al. "Judging LLM-as-a-judge with MT-Bench and Chatbot Arena." arXiv preprint arXiv:2306.05685 (2023).
>
> [2] Wang, Peiyi, et al. "Large language models are not fair evaluators." arXiv preprint arXiv:2305.17926 (2023).
>
> [3] Zeng, Zhiyuan, et al. "Evaluating large language models at evaluating instruction following." arXiv preprint arXiv:2310.07641 (2023).

---

> > ### Comment · Reviewer_RHzc · 2023-11-23
> > **Response to Authors**
> >
> > I thank the authors for their response. My concerns raised in my original review have been partially resolved (human judgments & stronger baselines). However, I do have several important remaining concerns:
> > * The human study (Section 4.4) is too small-scale with only 80 samples, and there is no inter-rater agreement rate reported, so we don't have an idea of the representativeness/reliability of the human study results. I still believe that enhancing the consistency rate is only one important aspect of LLM-evaluators, and there should be no sacrifice in the accuracy in this process (this ought to be validated with larger-scale and more rigorous human evaluations).
> > * Regarding the method design, I understand that the proposed method does not require additional forward passes to the LLM evaluators, but splitting the responses into pieces and merging them by segments still raises the concerns of hampering semantic coherence (since the LLMs have been trained on long and coherent text sequence completion tasks and have not been explicitly trained to evaluate split responses). There need to be additional studies to support that semantic coherence and long-range dependency could be preserved.
> > * For the generalization of the method, as I mentioned in my original review, and also reflected in other reviewers' comments, I'm still unconvinced that the method is very flexible in evaluating all types of open-ended generation scenarios. I don't think this issue could be addressed within the (short) rebuttal period but rather necessitate some rethinking & redesign of the method.
> >
> > Overall, I believe the paper could be (and likely should be) improved in various aspects. Nevertheless, considering that the authors won't have enough time to respond to my comments raised above and that they have addressed some of my initial concerns, I've raised my rating to 6 to not gatekeep the paper, though I'm more neutral than positive about the paper.

---

### Author Response · Authors · 2023-11-16
**Global response**

### Generalizability for Open-Ended Questions
To clarify, our framework is generally applicable to open-ended questions, as evidenced from three perspectives:
1. Diversity of Evaluated Questions: Our evaluation already included a broad spectrum of open-ended questions from eight primary categories, such as writing, reasoning, and roleplay, in our selected benchmark. Interested readers can refer to the benchmark's website at https://lmsys.org/vicuna_eval/ for an in-depth look at the open-ended nature of some responses in different categories, such as "roleplay-Q13" and "counterfactual-Q44".
2. Variety of LLMs Tested: The eight distinct LLMs tested in our study exhibited varying capabilities and preferences, further contributing to the diversity of responses.

3. Applicability to Open-Ended Responses: As demonstrated in Sections 4.2 and 4.4, our approach is not only applicable to point-to-point responses (as exemplified in Figure 1) but is also effective with more diverse responses. Importantly, our framework does not necessitate that the two responses being evaluated are of similar lengths. It works as long as the response contains content that can be segmented. In particular, we conduct three sets of additional result analyses below to emphasize this point.

**Response Statistics**

As pointed out by the reviewers, the generated results can differ significantly from each other.
To further explore this, we analyzed the statistical information of all responses, revealing substantial differences in response lengths within our benchmark dataset. The data is presented below.

|LLM|Max Len|Min Len|Avg Len|Standard Deviation|
|-|-|-|-|-|
|alpaca-13b|1149|6|508|222|
|bard|2652|151|1276|495|
|vicuna-7b|2598|266|1457|448|
|claude-v1|2392|94|1624|612|
|gpt-3.5|2218|193|1206|460|
|vicuna-13b|2441|212|1416|371|
|gpt-4|3842|201|2044|768|
|llama-13b|4827|9|757|895|
|gpt-3.5-short|365|26|152|58|

We observe that *the lengths of responses generated by the LLMs vary considerably*. For example, the maximum number of characters in the responses is 4,827 (llama-13b), while the minimum is just 6 characters (alpaca-13b).

**Relationship Between Answer Length and Inconsistency**

To further explore the relationship between answer length and inconsistency, we conducted an experiment using the collected judgment data. For this, "GPT-3.5" was used as the evaluator, analyzing 8 pairs of responses across three comparison forms.
The answers are categorized into 9 groups based on their length, with each group representing an 800-character interval. The results are presented below, with values below 2% of the total indicated by "-".

|# char (per 100 characters)|[0,8]|[8,16]|[16,24]|[24,32]|[32,40]|[40,48]|[48,56]|[56,64]|64+|
|-|-|-|-|-|-|-|-|-|-|
|% Incon Rate|-|-|26.89|23.02|31.84|39.01|42.73|55.45| - |

The table shows a generally positive correlation between answer length and inconsistency rate, with shorter answers tending to exhibit lower inconsistency rates. This finding suggests that position bias is less significant in shorter answers. When combined with the enhancement results detailed in Section 4.2, this leads to the conclusion that *the proposed framework is effective in handling responses of varying lengths*.

**Additional Experiments on Response Length**

However, our initial considerations did not account for a scenario where responses from one specific LLM are consistently and significantly shorter (e.g., 1/8th the length) than those from another. This is due to the expectation that LLMs under test are trained to generate responses adhering to given instructions, typically resulting in average response lengths of several hundred characters.

To determine if our framework remains applicable in such scenarios, we conducted an experiment with the following steps: (1) We instructed GPT-3.5 to shorten its responses while preserving as much meaning as possible, leading to a subset termed "gpt-3.5-short," which consisted of responses approximately 1/8th the length of the original ones. (2) We then used GPT-3.5 and GPT-4 as evaluators to compare "gpt-3.5-short" with "gpt-3.5" and "claude-v1" in exchanged orders, to assess consistency.

The results, shown below, indicate a 100% consistency rate (80/80) for both evaluators. This suggests that there is no inconsistency in this particular scenario, and therefore no alignment is needed. It means that *position bias is no longer a concern in such situations*. This finding aligns with previous studies [1,2], which noted that LLM evaluators prefer longer responses.

|||GPT-3.5|GPT-4|
|-|-|-|-|
|Model1|Model2|||
|gpt-3.5-short|gpt-3.5|100%|100%|
|gpt-3.5-short|claude-v1|100%|100%|

To summarize, we argue that our framework is applicable to open-ended questions, accommodating responses that vary significantly from each other, even in cases where one set of responses is systematically and markedly shorter than the other.

---

> ### Author Response · Authors · 2023-11-23
> **Second Global Response**
>
> Dear Reviewers,
>
> We thank you again for your constructive reviews and appreciate the time and effort you have invested in discussing with us.
>
> We would like to clarify one final point: our system, PORTIA, is specifically designed to address the **position bias** issue, rather than aligning the **reasoning capabilities** in LLM-based evaluators.
> Therefore, if an LLM evaluator lacks the capability to adequately judge a pair of semantically different responses, this falls outside the scope of PORTIA's intended purpose.
> PORTIA is solely focused on **rectifying inconsistencies** that arise when the order of a pair of semantically different responses is altered, leading to a change in the judgment of the LLM evaluator.
>
> In this context, we believe that PORTIA is generalizable to handling open-ended answers because, regardless of their semantic differences,
> **we can always mix** the two previously individual answers (as illustrated in Figure 1).
> Such a mixing operation effectively eliminates the position bias in pairwise LLM-based evaluation, as indicated in our evaluation data and supplementary experiments.
>
> Best regards,
>
> Anonymous Authors

---

### Meta-Review · Area_Chair_NUCA · 2023-12-04

**Metareview:**

The paper introduces a method to address position bias in LLM, meaning model preferring answer options presented earlier in-context. To address this, they propose a simple method that interweaves the outputs of answer options, which decreases model’s position bias. They evaluate this on many LMs (LLAMA2, GPTs, Claude2, etc) on MT-Bench benchmark dataset, showing the proposed methods improve consistency across the board. The reviewers have concerns with the generalizability of the method — whether this can be helpful for settings where answer options have different lengths / semantically different / etc. I agree that evaluating this on wider set of benchmark dataset would strengthen the paper, currently evaluation is only on 80 questions. I also think this overall algorithm (as noted by authors themselves in the response) only is effective/necessary in *lengthy* answer options setting, and such limitations of contribution should be presented clearly.

**Justification For Why Not Higher Score:**

Reviewer YB2D’s comment, that whether proposed method might make things more *consistent* but *harder* for LLM to evaluate output, is not addressed sufficiently in my opinion. Section 4.4 (human evaluation) can potentially address this better, but as the other reviewer pointed out, it’s small scale without proper protocols (e.g., inter annotator agreements). I don’t think you have to rely on human evaluation here — you can do automatic evaluation (whether the model will predict the correct answer more after PORTIA reordering?). I also agree with reviewer SWLn that the algorithm can be more clearly presented (many things in algorithm block is not properly defined).

**Justification For Why Not Lower Score:**

The motivation is clear and experimental results are clearly represented.

---

### Decision · Program_Chairs · 2024-01-16

Reject